# Intestinal AMPK modulation of microbiota mediates crosstalk with brown fat to control thermogenesis

Eryun Zhang [1,2], Lihua Jin[2], Yangmeng Wang[2], Jui Tu[2,3], Ruirong Zheng[1], Lili Ding[1,2], Zhipeng Fang [2], Mingjie Fan[2], Ismail Al-Abdullah[4], Rama Natarajan [2], Ke Ma[2], Zhengtao Wang[1], Arthur D. Riggs[2], Sarah C. Shuck [5], Li Yang [1✉] & Wendong Huang [2,3✉]

The energy-dissipating capacity of brown adipose tissue through thermogenesis can be targeted to improve energy balance. Mammalian 5′-AMP-activated protein kinase, a key nutrient sensor for maintaining cellular energy status, is a known therapeutic target in Type II diabetes. Despite its well-established roles in regulating glucose metabolism in various tissues, the functions of AMPK in the intestine remain largely unexplored. Here we show that AMPKα1 deficiency in the intestine results in weight gain and impaired glucose tolerance under high fat diet feeding, while metformin administration fails to ameliorate these metabolic disorders in intestinal AMPKα1 knockout mice. Further, AMPKα1 in the intestine communicates with brown adipose tissue to promote thermogenesis. Mechanistically, we uncover a link between intestinal AMPKα1 activation and BAT thermogenic regulation through modulating anti-microbial peptide-controlled gut microbiota and the metabolites. Our findings identify AMPKα1-mediated mechanisms of intestine-BAT communication that may partially underlie the therapeutic effects of metformin.

[1] Shanghai Key Laboratory of Compound Chinese Medicines and The Ministry of Education (MOE) Key Laboratory of Standardization of Chinese Medicines, Institute of Chinese Materia Medica, Shanghai University of Traditional Chinese Medicine, Shanghai, China. [2] Department of Diabetes Complications & Metabolism, Arthur Riggs Diabetes and Metabolism Research Institute, Beckman Research Institute, City of Hope National Medical Center, Duarte, CA, USA. [3] Irell & Manella Graduate School of Biological Science, City of Hope National Medical Center, Duarte, CA, USA. [4] Department of Translational Research & Cellular Therapeutics, Arthur Riggs Diabetes and Metabolism Research Institute, Beckman Research Institute, City of Hope National Medical Center, Duarte, CA, USA. [5] Department of Diabetes and Cancer Metabolism, Arthur Riggs Diabetes and Metabolism Research Institute, Beckman Research Institute, City of Hope National Medical Center, Duarte, CA, USA. ✉email: yl7@shutcm.edu.cn; whuang@coh.org

Obesity develops due to an imbalance between nutrient intake and energy expenditure, current treatments are limited in their ability to mitigate obesity and its complications[1]. Therefore, there is an urgent need to better understand the mechanisms governing nutrition and energy balance, with the long-term goal of developing novel strategies to treat obesity and associated diseases.

Mammalian 5′-AMP-activated protein kinase (AMPK) is a nutrient sensor that is essential for maintaining cellular energy homeostasis[2–4]. AMPK is expressed in several tissues, including the liver, brain, adipose tissues, skeletal muscle, and intestine[2,4]. When cellular energy is low, AMPK is activated by phosphorylation at residue T172 through the action of upstream kinases, such as LKB1[5]. The net effects of AMPK activation are stimulation of fatty acid (FA) oxidation and ketogenesis in the liver; FA oxidation and glucose uptake in skeletal muscle; and lipolysis and thermogenesis in adipocytes. AMPK activation also leads to the inhibition of cholesterol synthesis and lipogenesis in the liver; lipogenesis in adipocytes; and insulin secretion by pancreatic β-cells[3,6–9]. Given its critical roles in controlling energy homeostasis, AMPK has attracted widespread interest as a potential therapeutic target for metabolic diseases, including obesity, type 2 diabetes (T2D), and non-alcoholic fatty liver disease (NAFLD)[6,10].

Although AMPK has well-established roles in regulating energy and metabolism in many tissues, little attention has been paid to its role in the intestine, which is the first-line organ to process nutrients and is closely associated with diabetic postprandial dyslipidemia. It was reported that the improvements of glucose control by AMPK activators, including metformin, 5-aminoimidazole-4-carboxamide riboside (AICAR), resveratrol, and cucurbitacin B, are accompanied by the activation of AMPK signaling in the intestine[4,11–16], suggesting a significant contribution of intestinal AMPK in global glucose homeostasis. Notably, when small intestinal AMPK was virally knocked down in diabetic rodents, the glucose-lowering ability of acute metformin treatment was diminished by about 50%[11], suggesting an important contribution of intestinal AMPK activation to the therapeutic effects of metformin. However, the role of intestinal AMPK in regulating glucose homeostasis, as well as energy balance, remains largely unexplored.

To fill these gaps in knowledge regarding the action of intestinal AMPK, we generated an intestinal epithelium-specific AMPKα1 knockout mouse model (AMPKα1-IKO). Interestingly, we observed impaired BAT thermogenesis in the AMPKα1-IKO mice. Moreover, the gut microbiota profile of AMPKα1-IKO mice was markedly shifted compared to that of AMPKα1$^{fl/fl}$ control mice. Similarly, we identified significant differences in the serum levels of some bacterial metabolites in AMPKα1-IKO mice. Furthermore, expression of various anti-microbial peptides (AMPs) was significantly lower in the intestines of AMPKα1-IKO mice compared to AMPKα1$^{fl/fl}$ mice. Finally, we found that the beneficial metabolic effect of metformin was dependent on the intestinal AMPKα1. Taken together, this study identified AMPK-mediated mechanisms of intestine-BAT communication that may partially underlie the therapeutic effects of metformin.

## Results

### Intestinal AMPK regulates BAT thermogenesis
To address this current knowledge gap in our understanding of AMPK action, we genetically ablated AMPKα1, the predominant AMPK α subunit, selectively in the intestinal epithelial cells (IECs) (Supplementary Fig. 1a)[17,18]. On normal chow diet, the IEC-specific AMPK knockout (AMPKα1-IKO) mice did not exhibit an overt developmental or metabolic phenotypes as compared to AMPKα1$^{fl/fl}$

mice (Supplementary Fig. 1b–h). Surprisingly, however, BAT sections from AMPKα1-IKO mice displayed marked adipocyte hypertrophy with more abundant multi-locular lipid droplets than that of the controls (Fig. 1a, Supplementary Fig. 1i). Further analysis revealed consistent down-regulation of the thermogenic gene program, including key regulators UCP1 (uncoupling protein 1), ADRB3 (adrenoceptor beta 3), PGC-1α (peroxisome proliferator-activated receptor gamma coactivator 1-alpha), and Dio2 (iodothyronine deiodinase 2) (Fig. 1b). Consistent with the impaired thermogenic program, genes involved in mitochondrial function, fatty acid oxidation, and lipolysis pathways, were significantly attenuated in the BAT of AMPKα1-IKO mice (Fig. 1b). UCP1 protein level was markedly reduced in the BAT of AMPKα1-IKO mice than that of control AMPKα1$^{fl/fl}$ mice, both at room temperature and under cold exposure (Fig. 1c, Supplementary Fig. 1j). Further corroborating these findings, indirect calorimetry assay demonstrated a ~50% reduction of energy expenditure in intestinal AMPK-null mice at ambient temperature, as indicated by oxygen consumption (VO$_2$) and carbon dioxide production (VCO$_2$) rate (Fig. 1d, e, Supplementary Fig. 1k–n). Consistent with the attenuated thermogenic program leading to lower heat production, AMPKα1-IKO mice displayed lower rectal temperatures than AMPKα1$^{fl/fl}$ mice under cold exposure (Fig. 1f).

We next tested the response of intestinal AMPK in regulating energy balance when challenged by over-nutrition using a diet-induced obese (DIO) mouse model. Despite similar food intake (Supplementary Fig. 1o) on the high-fat diet (HFD), AMPKα1-IKO mice gained significantly more weight than AMPKα1$^{fl/fl}$ mice (Fig. 1g, Supplementary Fig. 1p, q), and developed markedly impaired glucose and insulin tolerance as compared to AMPKα1$^{fl/fl}$ mice (Fig. 1h–k). Moreover, the glucoregulatory effect in AMPKα1-IKO mice is weight-independent (Supplementary Fig. 1r). Consistent with this, expression of genes involved in hepatic gluconeogenesis were significantly increased in AMPKα1-IKO mice (Supplementary Fig. 1s) while hepatic lipid accumulation did not change (Supplementary Fig. 1t–u), suggesting a key role of intestinal AMPK in regulating hepatic gluconeogenesis. In line with previous observations, attenuated energy balance and glucose metabolism were accompanied by suppression of thermogenic program and UCP1 protein expression in BAT (Fig. 1l, m, Supplementary Fig. 1v), together with markedly lower energy expenditure (Fig. 1n, o, Supplementary Fig. 1w–z). These results indicate a protective function of intestinal AMPK against over-nutrition through maintenance of BAT thermogenic regulation. In addition, we examined the phenotypes in muscle and inguinal WAT (iWAT), however we did not observe significant changes in the morphology of these tissues, as well as the expression of genes involved in thermogenesis and glucose metabolism (Supplementary Fig. 2a–f)

### Gut microbiota and the metabolites mediate crosstalk between intestinal AMPK and BAT
Recent reports indicate that changes in gut microbiota can modulate BAT thermogenesis and whole-body energy expenditure[19–21]. Intriguingly, as oral administration of AMPK activators metformin and berberine alters gut microbiota composition[22], intestinal AMPK may mediate crosstalk with BAT through regulatory effects on gut microbiota profile. We thus profiled gut microbiota in AMPK α1-IKO and AMPKα1$^{fl/fl}$ mice on chow diet. Significantly altered gut microbiota in AMPKα1-IKO mice was observed as compared to that of AMPKα1$^{fl/fl}$ mice (Fig. 2a, Supplementary Fig. 3a–d), including the abundance of bacteria species known to modulate glucose, lipid, and energy metabolism (Fig. 3b). Thus, gut microbiota may mediate specific functions of intestinal AMPK in metabolism. We

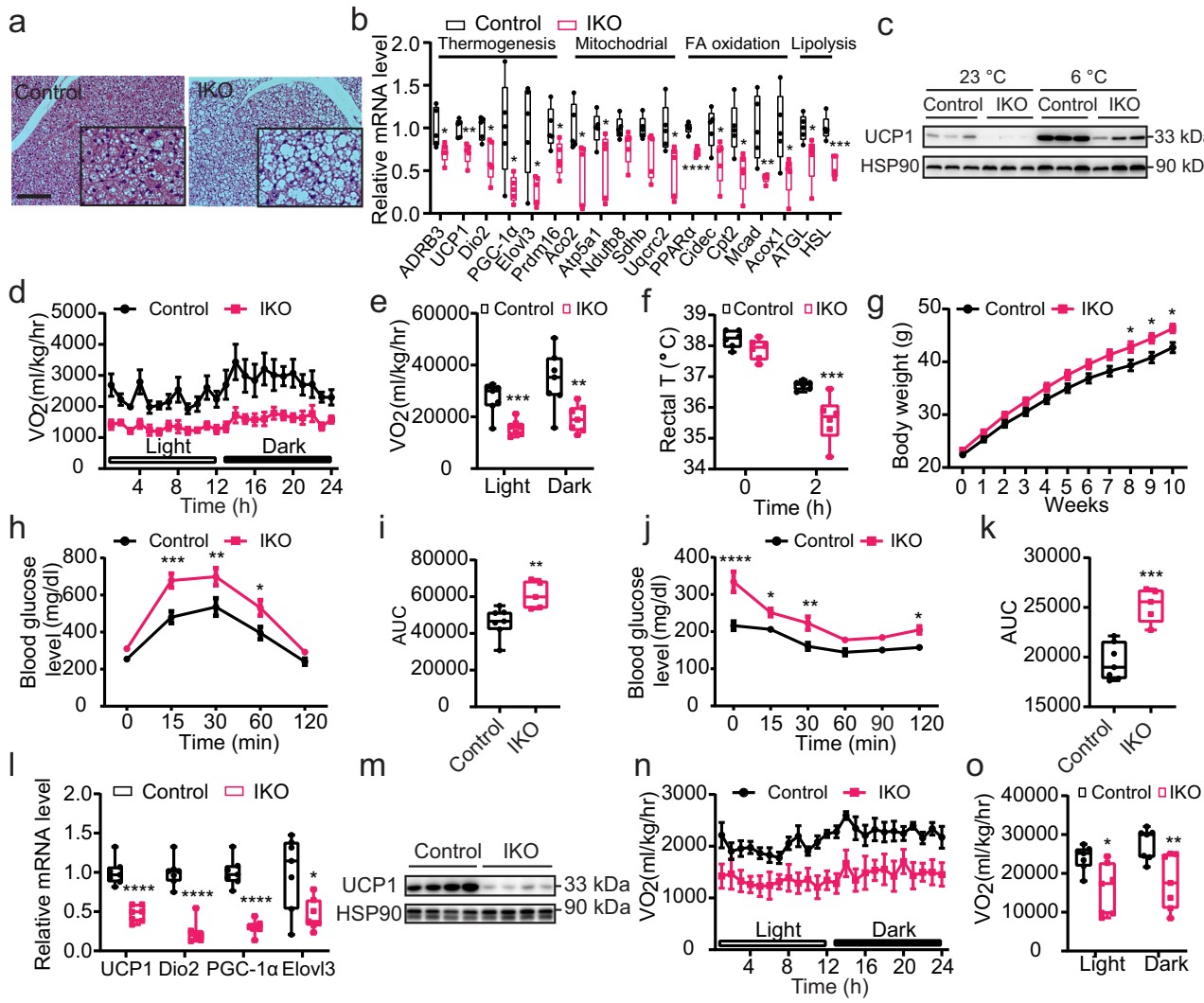

**Fig. 1 Intestinal epithelial AMPK knockout mice have impaired BAT function.** For **a–f**, AMPKα1[fl/fl] (Control) and AMPKα1-IKO mice were fed chow diet. **a** Representative images of H&E-stained BAT sections. Scale bar = 100 μm. **b** Relative mRNA levels of genes in BAT (n = 5 biologically independent 0.0495, 0.0271, <0.0001, 0.0364, 0.0152, 0.0093, 0.0329, 0.0328, 0.0006. **c** Western blot analysis of UCP1 protein levels in BAT before and after cold exposure (6 °C) for 1 week. **d, e** VO2 of mice over 24 h (n = 7 biologically independent samples). P value: 0.0007, 0.0075. **f** Rectal temperatures of mice exposed to 6 °C for 2 h (n = 6 biologically independent samples). P value: 0.0004. For **g–o**, mice were fed HFD for 10–12 weeks. **g** Body weights of mice during the HFD feeding period. Mice were fed HFD from 6-week-old (n = 20 biologically independent samples for control group, n = 16 biologically independent samples for IKO group). P value: 0.0326, 0.0197, 0.0209. **h, i** Glucose tolerance test results at 10 weeks of HFD feeding (n = 7 biologically independent samples for control group, n = 5 group). P value for **h**: 0.0004, 0.0047, 0.0255. P value for **i**: 0.0088. **j, k** Insulin tolerance test results at 11 weeks of HFD feeding (n = 7 biologically independent samples for control group, n = 5 biologically independent samples for IKO group). P value for **j**: <0.0001, 0.0301, 0.0011, 0.0225. P value for **k**: 0.0003. **l** Relative mRNA levels of thermogenesis genes in BAT (n = 7 biologically independent samples). P value: <0.0001, <0.0001, <0.0001, 0.0165. **m** UCP1 protein levels in BAT. **n, o** VO2 of mice over 24 h (n = 7 biologically independent samples). P value: 0.0119, 0.0079. Values are means ± s.e.m. for **d, g, h, j, n**. The boxplot elements (for **b, e, f, i, k, l**, and **o**) are defined as following: center line, median; box limits, upper and lower quartiles; whiskers, 1.5 × interquartile range. *P < 0.05, **P < 0.01, ***P < 0.001 and ****P < 0.0001 cf. control mice by two-tailed Student's t-tests (for **b, e, i, k, l, o**) and two-way ANOVA with Tukey's post-hoc tests for **f–h, o**. Source data are provided as a Source Data file.

examined the changes of gut microbiota at the family level and found that the relative abundance of families *Muribaculaceae*, *Prevotellaceae* and *Mycoplasmataceae* were significantly higher, while the relative abundance of family *Lachnospiraceae* was significantly lower in AMPK α1 IKO mice (Supplementary Fig. 3b). Interestingly, the relative level of *Lachnospiraceae* family has been reported to be induced by cold stimulation in WT mice[23], suggesting that *Lachnospiraceae* family maybe involved in the intestinal AMPK-mediated energy expenditure.

To further confirm that gut microbiota mediates intestinal AMPK action in metabolic homeostasis, we performed fecal microbiota transplantation (FMT) using microbiota harvested

from AMPKα1-IKO mice. FMT from AMPKα1-IKO, as compared to FMT from AMPKα1[fl/fl] mice, led to markedly increased lipid accumulation in BAT sections as indicated by more abundant lipid droplets (Fig. 2c, Supplementary Fig. 4a). Consistently, the thermogenic gene program genes and related pathways of BAT function, together with UCP1 protein, were all significantly reduced in the BAT of mice received FMT from AMPKα1-IKO (Fig. 2d, e, Supplementary Fig. 4b). Further corroborating these observations, when the recipient mice were exposed to 6 °C for 2 h, mice that received FMT from AMPKα1-IKO had significantly lower rectal temperature (Fig. 2f), suggesting impaired heat production. This result was further supported

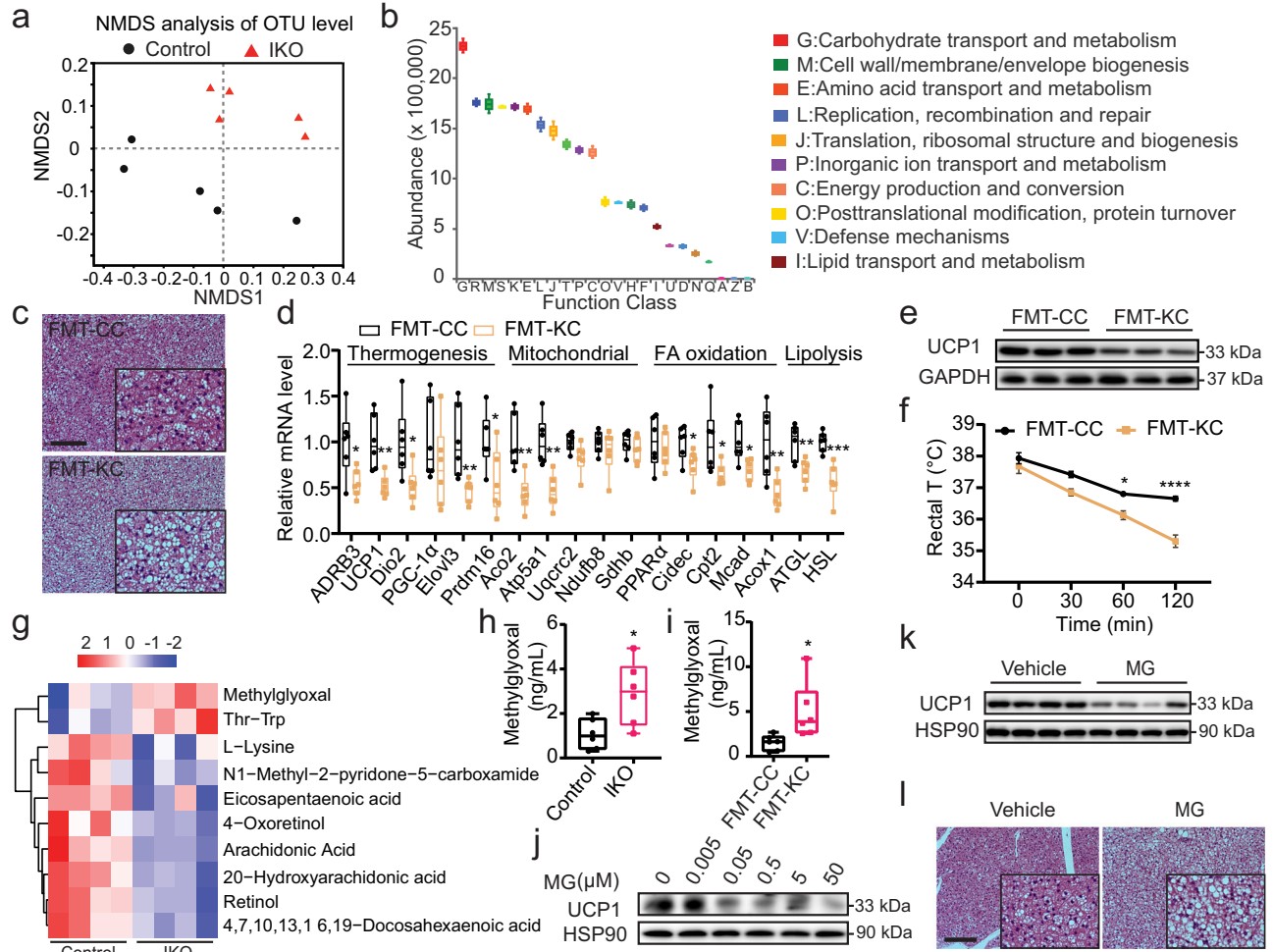

**Fig. 2 Intestinal AMPK remotely regulates BAT function by modulating gut microbiota and their metabolites. a** Non-metric multidimensional scaling (NMDS) analysis of operational taxonomic unit (OTU) levels in AMPKα1fl/fl (Control) and AMPKα1-IKO mice fed chow diet (n = 5 biologically independent samples). **b** Functional classification of microbiota with differential abundance in AMPKα1-IKO mice compared to AMPKα1fl/fl mice, based on the Clusters of Orthologous Groups (COGs) database (n = 5 biologically independent samples). **c** Representative images of H&E-stained BAT sections from FMT recipient mice. Scale bar = 100 μm. FMT-CC, FMT from AMPKα1fl/fl mice to WT mice; FMT-KC, FMT from AMPKα1-IKO mice to WT mice. **d** Relative mRNA levels of genes expressed in the BAT of recipient mice (n = 6 biologically independent samples). P value: 0.0171, 0.0041, 0.0203, 0.0097, 0.0380, 0.0025, 0.0018, 0.0357, 0.0371, 0.0121, 0.0080, 0.0080, 0.0007. **e** Representative Western blot analysis of UCP1 protein levels in the BAT of recipient mice. **f** Rectal temperatures of recipient mice exposed to 6 °C for 2 h (n = 6 biologically independent samples for FMT-CC group and n = 8 biologically independent samples for FMT-KC group). P value: 0.0152, <0.0001. **g** Metabolomic analysis of serum samples from DIO Control and AMPKα1-IKO mice. **h, i** Serum methylglyoxal levels in DIO mice (**h**) and FMT recipient mice (**i**), measured using ELISA (n = 6 biologically independent samples). P value for **h**: 0.0172, P value for **i**: 0.0266. **j** Representative Western blot analysis of UCP1 protein expression in HIB1B cells treated with methylglyoxal (MG) for 24 h. **k** UCP1 protein expression in the BAT of WT mice after intraperitoneally (i.p.) with methylglyoxal (50 mg/kg) once daily for 14 days. **l** Representative images of H&E-stained BAT sections of mice treated with methylglyoxal. Scale bar = 100 μm. Values are means ± s.e.m. for **b**, **f**. The boxplot elements (for **d**, **h**, **i**) are defined as following: center line, median; box limits, upper and lower quartiles; whiskers, 1.5 × interquartile range. *P < 0.05, **P < 0.01, ***P < 0.001 and ****P < 0.0001 by two-tailed Student's t-tests for **d**, **h**, **j** and two-way ANOVA with Tukey's post-hoc tests for (**f**). Source data are provided as a Source Data file.

by the functional gene profile of BAT of mice under cold exposure (Supplementary Fig. 4c–e). Importantly, the impaired thermogenesis in AMPKα1-IKO mice can be rescued by using microbiota from WT mice (Supplementary Fig. 4f–k). Together, these results indicate that gut microbiota from AMPKα1-IKO mice is sufficient to impair BAT thermogenesis, suggesting that intestinal AMPK may confer its regulation of BAT through gut microbiota.

Gut microbiota primarily regulate BAT thermogenic functions through their metabolites[24–27]. we thus conducted a metabolomic analysis of serum samples from DIO AMPKα1-IKO and AMPKfl/fl. Indeed, intestinal AMPK deficiency led to significantly altered

serum metabolites (Fig. 2g). Methylglyoxal, one of the altered metabolites produced by intestinal bacteria and host metabolism, was of particular interest due to its high levels in the patients with diabetes and known involvement in development of T2D[28–32]. We further confirmed the significantly higher methylglyoxal levels in serum, BAT and fecal samples of AMPKα1-IKO mice compared to AMPKα1fl/fl mice (Fig. 2h, Supplementary Figs. 4l and 5). Notably, FMT from AMPKα1-IKO mice was able to increase methylglyoxal levels in recipient mice in not only fecal samples but also serum (Fig. 2i, Supplementary Fig. 4m), suggesting that gut microbiota altered by AMPKα1-IKO contributed greatly to the increased circulating methylglyoxal level. We then directly tested the

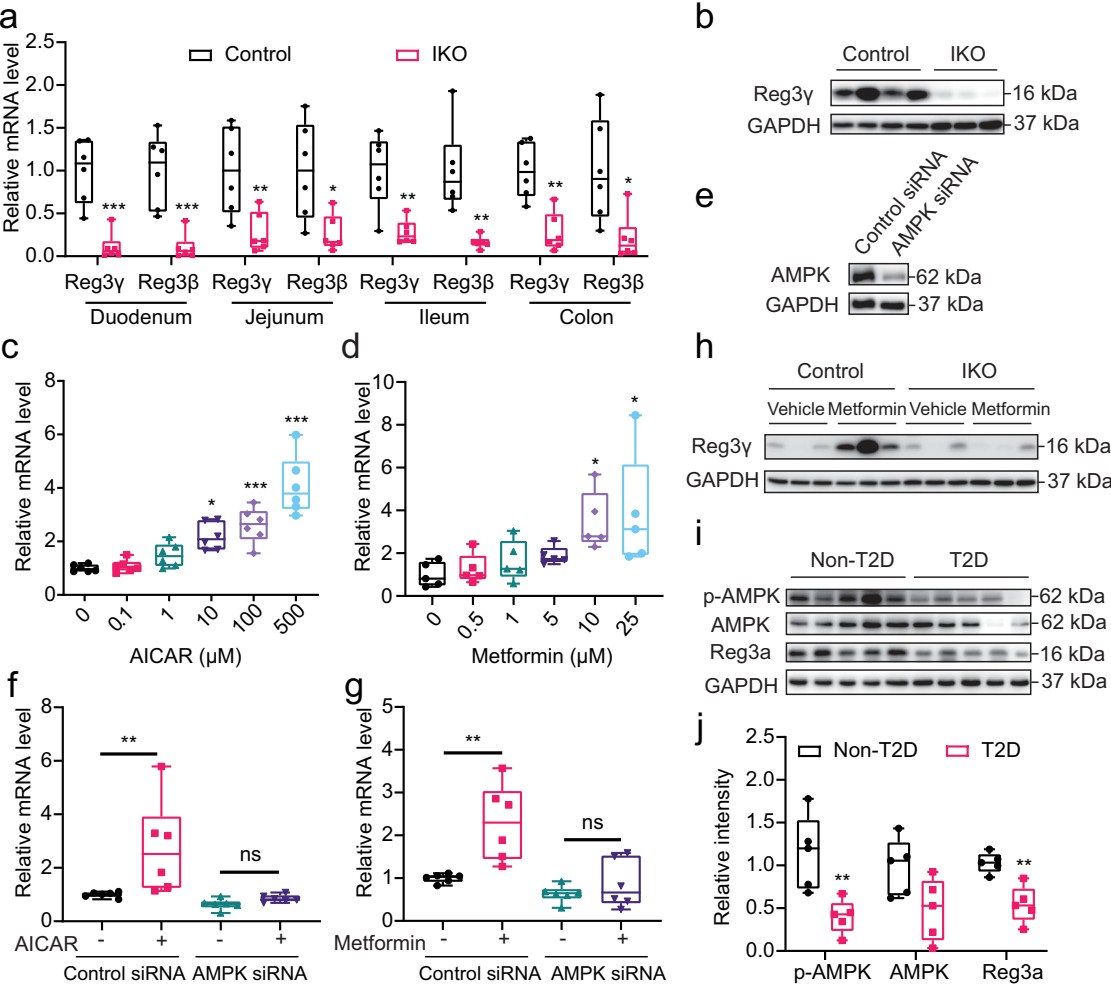

**Fig. 3 Intestinal AMPKα1 regulates the AMP Reg3. a** Relative mRNA levels of Reg3γ and Reg3β in the duodenum, jejunum, ileum, and colon of AMPKα1$^{fl/fl}$ (Control) and AMPKα1-IKO mice fed chow diet ($n = 6$ biologically independent samples). P value: 0.0003, 0.0007, 0.0095, 0.0143, 0.0029, 0.0022, 0.0011, 0.0143. **b** Representative Reg3γ protein levels in the jejunum of mice fed chow diet. **c, d** Relative mRNA levels of Reg3α in HT-29 cells treated with gradient concentrations of AICAR (**c**, $n = 6$ biologically independent samples, P value: 0.0078, 0.0004, <0.0001) and metformin (**d**, $n = 5$ biologically independent samples, P value: 0.0315, 0.0120). **e** Western blot analysis of AMPK protein in HT-29 cells treated with control siRNA or AMPK siRNA. **f, g** Relative mRNA levels of Reg3α in AMPK knockdown HT-29 cells treated with 100 μM of AICAR (**f**) or 10 μM of metformin (**g**) for 24 h. $n = 6$ biologically independent samples. P value for **f**: 0.0128. P value for **g**: 0.0023. **h** Representative Reg3γ protein levels in WT DIO mice treated with metformin. **i** Western blot analysis of total AMPK, phosphorylated AMPK (p-AMPK), and Reg3α in duodenal mucosa samples from patients with obesity and T2D. **j** The quantitated densities of these Western blot bands are also shown ($n = 5$ biologically independent samples). P value: 0.0089, 0.0023. The boxplot elements (for **a, c, d, f, g, j**) are defined as following: center line, median; box limits, upper and lower quartiles; whiskers, 1.5 × interquartile range. *$P < 0.05$, **$P < 0.01$, ***$P < 0.001$ and ****$P < 0.0001$ by two-tailed Student's t-test (for **a**, cf. Control mice; for **j**, cf. non-T2D human samples), and one- or two-way ANOVA with Tukey's post-hoc tests (for **c, d**, cf. vehicle-treated cells; for **f, g**, cf. vehicle-treated cells infected with control siRNA). Source data are provided as a Source Data file.

potential of methylglyoxal on modulating thermogenic regulation in brown adipocytes using the HIB1B cell line. Indeed, methylglyoxal induced significant suppression of UCP1 protein in a dose-dependent manner in differentiated HIB1B cells (Fig. 2j). Various pathways related to BAT function was also reduced by methylglyoxal treatment (Supplementary Fig. 6a). Furthermore, we show that administration of methylglyoxal in mice by intraperitoneal injections reduced UCP1 expression in BAT (Fig. 2k, Supplementary Fig. 6b) and induced lipid accumulation in brown adipocytes (Fig. 2l, Supplementary Fig. 6c), suggesting direct effect of methylglyoxal on lowering BAT activity both in brown adipocyte and in vivo. Viability of HIB1B cells treated with gradient doses of methylglyoxal (Supplementary Fig. 6d) and the protein levels of cell death markers Bcl2 and cleaved-caspase 3 in the BAT of methylglyoxal-treated mice (Supplementary Fig. 6e) indicated that the effects of methylglyoxal on UCP1 levels were not due to cell

death. In aggregate, these data indicate that gut microbiota metabolites, particularly methylglyoxal, could mediate the crosstalk between intestinal AMPK action and BAT thermogenic regulation.

**Intestinal AMPK regulates expression of AMPs.** AMPs, the gut-bacterial-derived peptides, have been shown to effectively kill pathogens[33] to maintain an effective barrier against microbes on intestinal epithelium. Considering the critical role of AMPs in determining microbiota composition and markedly altered microbiota profile of AMPKα1-IKO mice, we postulate that AMPK action in the intestine may directly modulate AMP expression. Interestingly, we found markedly reduced mRNA and protein levels of Reg3γ, a key AMP, in the intestinal tissue of AMPKα1-IKO mice than in that of AMPKα1$^{fl/fl}$ mice (Fig. 3a, b, Supplementary Figs. 7 and 8a). Moreover, expression of several

other AMPs, including Reg3β, resistin-like molecule β (RELMβ), matrix metallopeptidase 7 (MMP7), and α-defensin (Defa), were similarly reduced by loss of AMPK (Supplementary Figs. 7 and 8b). In contrast, treatment by AMPK activator AICAR directly induced AMP expression in human epithelial HT-29 cells (Fig. 3c, Supplementary Fig. 8c). Reg3γ in the gut can directly target bacteria by attacking peptidoglycan carbohydrate backbone of both Gram-positive and Gram-negative species[34,35]. We therefore chose to further examine Reg3γ regulation by AMPK as a key representative AMP. AMPK activators AICAR and metformin-induced Reg3α transcript (the human homolog of mouse Reg3γ) in HT-29 cells in a dose-dependent manner (Fig. 3c, d); whereas AMPK knockdown impaired these effects (Fig. 3e–g). Furthermore, metformin strongly restored Reg3γ expression in the small intestines of AMPKα1fl/fl mice but not AMPKα1-IKO mice (Fig. 3h, Supplementary Fig. 8d), providing support that Reg3 expression is directly controlled by AMPK signaling. Most importantly, in the patients with obesity and T2D, we observed significantly reduced Reg3α expression, together with evidence of attenuated intestinal AMPK T172 phosphorylation suggestive of its activity (Fig. 3i, j). Taken together, these findings suggest that intestinal AMPK may modulate gut microbiota composition through regulation of AMPs, such as Reg3γ, in both rodents and humans.

**Intestinal AMPK is required for the metabolic benefits of metformin in HFD.** Metformin is a first-line anti-diabetes drug[36–38] and accumulating evidence indicate its action in the gut is important to glucose control[11,39–41]. However, the underlying molecular mechanisms of this contribution remain unclear. Strikingly, as shown in Fig. 4a and f, despite similar food intake (Supplementary Fig. 9a, b), the effect of metformin on inducing body weight loss in DIO-AMPKα1fl/fl mice (Fig. 4a) was largely abolished in the AMPKα1-IKO (Fig. 4f). Notably, similar reduction of metformin effect on improving glucose tolerance (Fig. 4b and f), hepatic lipid accumulation (Fig. 4c and h, Supplementary Fig. 9c), and reducing fasting serum levels of triglycerides, total cholesterol and insulin (Fig. 4d and h, Supplementary Fig. 9d), were also observed. Although the glucoregulatory effect of metformin in WT mice was weight-independent, intestinal AMPK deficiency disabled metformin to further increase glucose tolerance (Supplementary Fig. 9e, f). These results indicated that the metabolic actions of metformin are dependent on intestinal AMPK.

Although AMPK is expressed in BAT and metformin can enhance AMPK activity in BAT to activate brown fat thermogenic activity[9,42,43], whether intestinal AMPK could contribute to the effects of metformin in BAT is yet to be tested. We found that metformin reduced lipid accumulation and increased UCP1 level in the BAT of AMPKfl/fl, but not AMPKα1-IKO mice (Fig. 4k, l, Supplementary Fig. 9g), supporting the idea that AMPK action in the gut is required for metformin regulation of BAT function. Interestingly, prior studies indicate that metformin alters the gut microbiota of the patients with obesity or diabetes[44–46], which may contribute to the therapeutic effects of metformin. In line with these findings, we found that metformin treatment significantly altered the relative amount of several bacterial genera in AMPKα1fl/fl (Fig. 4m, Supplementary Fig. 9i, j), such as the increased levels of *Bacteroides*[47], *Parabacteroides*[48–50], *Alistipes*[51,52], *Akkermansia*[53,54], *Parasutterella*[22] and *Klebsiella*[22,55], and decreased levels of *Blautia*[53], *Ruminiclostridium*[56], *Lachnospiraceae_NK4A136_group*[57], *Oscillibacter*[58], *Bilophila*[59], *Desulfovibrio*[49,57], *Streptococcus*[60], *Roseburia*[22,55], *Lachnoclostridium*[49] and *Anaerotruncus*[53] that are consistent with previous reports. Interestingly, most of them are changed in an intestinal AMPK-dependent

manner (Fig. 4m), suggesting that the metformin-controlled alteration of gut microbiota depends on intestinal AMPK. As expected, metformin induced Reg3γ expression in the intestines of AMPKα1fl/fl but not AMPKα1-IKO mice (Fig. 3h, Supplementary Fig. 9h). Further, the circulating methylglyoxal levels were significantly decreased in the metformin-treated AMPKα1fl/fl but not AMPKα1-IKO mice (Fig. 4e, j). Taken together, these results suggest that the metabolic effects of AMPK-activating metformin is dependent on AMPK function in the gut, likely through an intestinal AMPK-AMP-gut microbiota link.

## Discussion
The role of AMPK in cellular energy homeostasis is highly essential and conserved in eukaryotes. The intestine plays a central role in nutrient sensing, absorption, and metabolism. Our current study reveals that the crosstalk between intestinal AMPK and BAT regulates thermogenesis and metabolism through modulating AMP-controlled gut microbiota and the metabolites.

Previous reports have demonstrated that hepatic AMPK is required to mediate the anti-hyperglycemic effects of metformin[36,61]. However, there are also evidence showing that metformin inhibits hepatic gluconeogenesis in mice is independently of the LKB1/AMPK pathway[37,62] in the liver. On the other hand, accumulating evidence indicate that gut may be another critical site for metformin to exhibit its therapeutic effects. For instance, the peak of metformin concentrations in the jejunum has been reported at 30–300 times greater than plasma concentrations[63], indicating that most of the metformin administered by gavage distributes to the intestine. Moreover, delayed-release metformin, which is formulated to deliver the drug to target the lower bowel, was more effective at lowering fasting plasma glucose than currently available metformin[64]. In the gut, metformin can stimulate glucagon-like peptide-1 (GLP-1) release[11]. Metformin is also involved in modulating gut microbiota that can regulate metabolism[44,45]. Importantly, when small intestinal AMPK was virally knocked-down in diabetic rodents, the glucose-lowering ability of acute metformin treatment was diminished by about 50%[11]. Together, these results suggest a critical contribution of intestinal AMPK to the therapeutic effect of metformin, and might also explain the less extent to which hepatic AMPK deficiency impaired the therapeutic effects of metformin than that caused by intestinal AMPK knockout.

Overall, our findings provide a new mechanistic underpinning of AMPK function in metabolic regulation, which may shed light on the ongoing large clinical trials of metformin, as well as general drug discoveries utilizing enteral drug administration to treat metabolic diseases.

## Methods
The research complies with all relevant ethical regulations, and was approved by the Institutional Review Board of City of Hope (IRB No. 01046), and the City of Hope Institutional Animal Care and Use Committee (IACUC No. 14031).

**Clinical samples.** Duodenal mucosa specimens were collected from patients with obesity and T2D at the Southern California Islet Cell Resource Center at City of Hope. In brief, subjects were divided into patients with obesity and diabetes (n = 5) and patients without obesity and diabetes (n = 5). Obesity was defined as a body mass index of 30 or greater, and T2D was defined according to the stringent HbA1c guidelines established by the American Diabetes Association (T2D: HbA1c > 6.5%, approximately equivalent to 7.8 mM blood glucose) (Supplementary Table 1). All participants provided written informed consent, and ethical approval for this study was granted by the Institutional Review Board of City of Hope (IRB No. 01046).

**Animals.** Wild-type (WT) C57BL/6J, C57BL/6-AMPKa1flox/flox mice (#014141), and B6. Cg-Tg (Vil-cre) 1000Gum/J (#021504) were purchased from Jackson Laboratory (Bar Harbor, ME). AMPK α1-IKO mice were generated by cross-breeding C57BL/6-AMPKa1flox/flox mice with B6. Cg-Tg (Vil-cre) 1000Gum/J. All animal procedures were approved by the City of Hope Institutional Animal Care

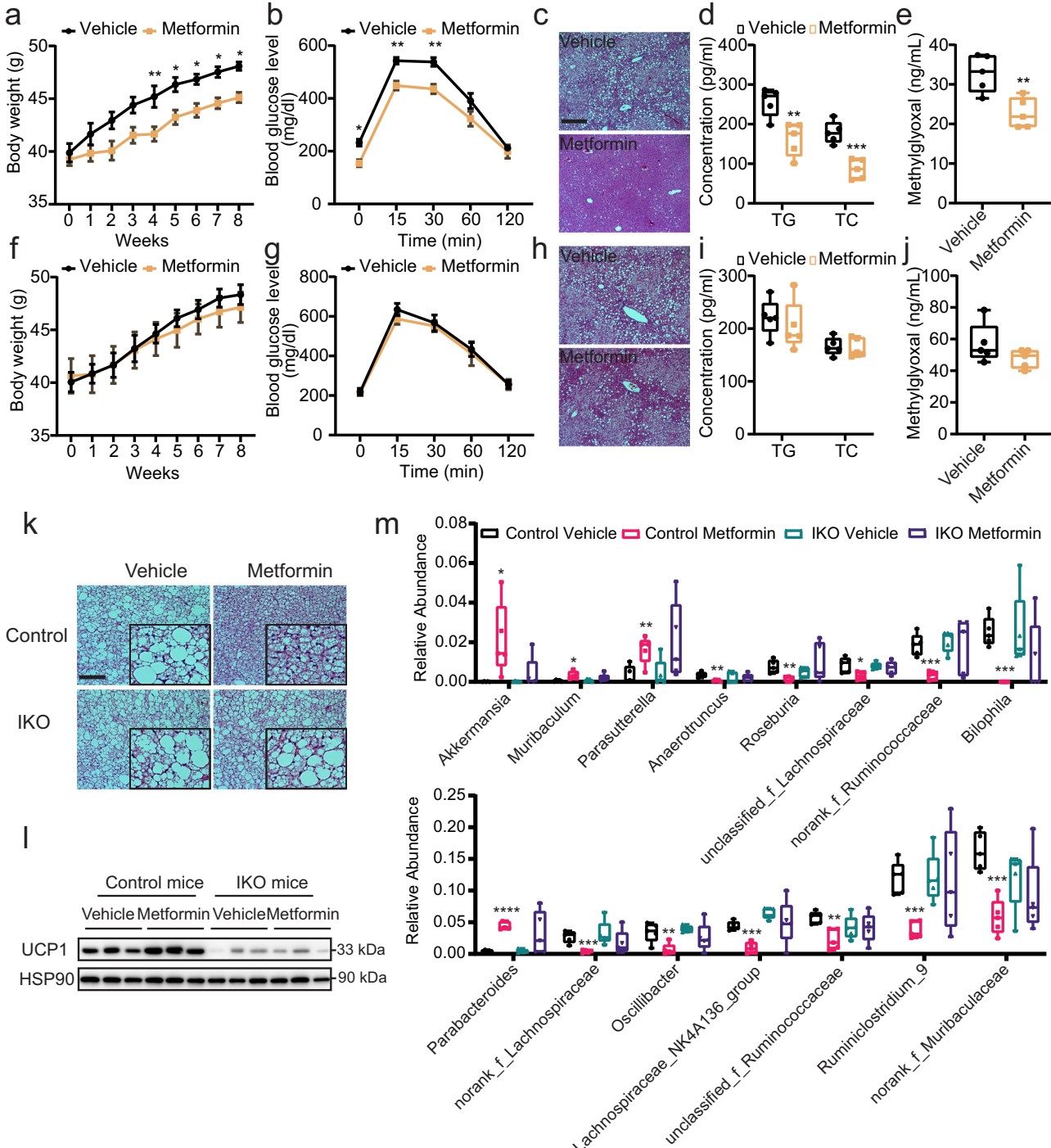

**Fig. 4 The therapeutic effects of metformin on metabolic disorders depend on intestinal AMPK.** DIO AMPKα1fl/fl (Control; **a–d**) and AMPKα1-IKO (**e–h**) mice were orally garaged with metformin (100 mg/kg) once daily for 8 weeks. **a, f** Body weights of mice during metformin treatment ($n = 7$ biologically independent samples). $P$ value for **a**: 0.0066, 0.0264, 0.0417, 0.0400, 0.0384. **b, g** Glucose tolerance test results at 8 weeks of metformin administration ($n = 7$ biologically independent samples). $P$ value for **b**: 0.0308, 0.0055, 0.0022. **c, h** Representative images of H&E-stained liver sections. Scale bar = 100 μm. **d, i** Fasting serum levels of triglycerides (TG) and total cholesterol (TC) ($n = 5$ biologically independent samples). $P$ value for d: 0.0054, 0.0006. **e, j** Serum methylglyoxal levels measured by LC/MS–MS methods ($n = 5$ biologically independent samples). $P$ value for **e**: 0.0061. **k** Representative images of H&E-stained BAT sections, Scale bar = 100 μm. **l** Representative UCP1 protein expression in BAT. **m** Relative abundance of microbiota at the genus level ($n = 5$ biologically independent samples). $P$ value for the upper panel: 0.0313, 0.0173, 0.0081, 0.0016, 0.0049, 0.0193, 0.0004, 0.0001. $P$ value for the lower panel: <0.0001, 0.0007, 0.0091, 0.0001, 0.0032, 0.0002, 0.0005. Values are means ± s.e.m. for **a, b** and **f, g**. The boxplot elements are defined as following: center line, median; box limits, upper and lower quartiles; whiskers, 1.5 × interquartile range for **d, e, i, j**, and **m**. *$P < 0.05$, **$P < 0.01$, ***$P < 0.001$ and ****$P < 0.0001$ cf. vehicle-treated mice by two-tailed Student's $t$-test (for **d, e, i, j, m**), and two-way ANOVA with Tukey's post-hoc tests (for **a, b, f, g**). Source data are provided as a Source Data file.

and Use Committee (IACUC No. 14031) and conducted in accordance with the National Institutes of Health Guidelines for the Care and Use of Laboratory Animals. 8–12-week-old male mice were used in the animal experiments (unless otherwise indicated). Mice were housed in a temperature (22–23 °C) and light-controlled vivarium with free access to water and normal chow diet (17% kcal fat; Diet 8640, Harlan Teklad, Madison, WI) or high-fat diet (HFD, 60% kcal fat; D12492, Research Diets, New Brunswick, NJ). HFD was started at 6 weeks of age and continued for 12 weeks. All treatment groups were weight-matched and randomized to treatment groups at the initiation of each experiment. Cold exposure was performed at 6 °C in a light- and humidity-controlled (40%) climatic chamber under specific-pathogen-free (SPF) conditions. To assess whole-body energy metabolism, mice were placed into metabolic cages and acclimatized for approximately 48 h. Food and fluid intake, ambient locomotor activity, $O_2$ consumption ($VO_2$), $CO_2$ output ($VCO_2$), respiratory exchange ratio (RER), and energy expenditure were measured using a Comprehensive Laboratory Animal Monitoring System (Columbus Instruments). Body composition was measured using magnetic resonance imaging (EchoMRI, Houston, TX). To investigate the effects of methylglyoxal on BAT activity, wild-type mice were treated with intraperitoneal (i.p.) injections of methylglyoxal (50 mg/kg) once daily for 14 days. For chronic metformin studies, 6-week-old male AMPKα1$^{fl/fl}$ and AMPKα1-IKO mice were switched from ND chow to HFD for 6 weeks. Then, they were weight-matched and divided into two groups to receive vehicle treatment (tap water) or metformin by oral gavage (100 mg/kg) once daily for 8 weeks.

**Metabolic phenotype and blood measurements**. To test glucose tolerance, mice were fasted for 16 h before i.p. injection of glucose solution (1.5 mg/g body weight). Blood glucose levels were measured at baseline and at 15, 30, 60, and 120 min post-injection using an Accu-Chek glucometer (Roche Diagnostics, France). To test insulin tolerance, mice were fasted for 4 h before i.p. injection of insulin (0.75 U/kg body weight). Blood glucose levels were measured at baseline and at 15, 30, 60, and 120 min post-injection. Serum methylglyoxal levels were determined using a Methylglyoxal ELISA Kit (abcam, Cambridge, MA). Serum insulin was assessed after a 12 h of fasting using a Mouse Insulin ELISA Kit (Millipore Sigma, Burlington, MA) according to the manufacture's instructions. Plasma triglyceride and total cholesterol levels were determined using LabAssay Triglyceride and Cholesterol Kits (Wako Chemicals, Osaka, Japan), respectively.

**Gut microbiota profiling**. Bacterial DNA was extracted from the cecal contents of mice using a QIAamp DNA Stool Mini Kit (Qiagen). The V4 region of the bacterial 16S rRNA gene was amplified by triplicate PCR (F515/R806) using barcoded fusion primers. Samples were pooled in sets of a maximum of 96 samples in equal quantities. Paired-end sequencing of the amplicon library was performed on the Illumina MiSeq 300-bp paired-end platform. A multi-step bioinformatics analysis performed using QIIME 1.9.1 software included filtering raw fastq files for primer and adaptor dimer sequences, removing contaminating host sequences and chimeric sequences, clustering sequences into operational taxonomic units (OTUs) using the open reference OTU calling method with the greengenes 16S reference, and calculating alpha and beta diversity metrics. Non-metric multidimensional scaling (NMDS) is an ordination technique that aims to discover data patterns in N-dimensional space. It represents the major variation among objects in a reduced dimensional space. Linear discriminant analysis (LDA) of effect size (LEfSe) and the cladogram method were used for microbial biomarker discovery. Functional classification of microbiota was predicted using the PICRUSt (Phylogenetic Investigation of Communities by Reconstruction of Unobserved States) software package, based on the Clusters of Orthologous Groups (COGs) database. The data were analyzed on the free online Majorbio Cloud Platform (www.majorbio.com).

**Fecal microbiota transplantation**. FMT was performed as described in a previous study[65]. In brief, 8-week-old WT recipient mice were treated with a cocktail of antibiotics, including metronidazole, vancomycin, neomycin, and ampicillin to deplete their gut flora. First, they were administered the antibiotic cocktail in drinking water (per 1 l of water: 1 g metronidazole, 500 mg vancomycin, 1 g neomycin, and 1 g ampicillin) for one week. Then, they were orally gavaged with 200 μl of the antibiotic cocktail (per 1 ml of water: 0.2 g metronidazole, 0.1 g vancomycin, 0.2 g neomycin, and 0.2 g ampicillin) once daily for another week. Fresh feces were collected from donor mice under SPF conditions. Feces samples were weighed and diluted with 1 ml of saline per 0.1 g of stool, steeped for about 15 min, shaken, and then centrifuged at $800 \times g$ for 3 min. The supernatant was obtained for gavage once daily for 4 weeks.

**Histology analysis**. Tissues were fixed in 10% formalin and were paraffin-embedded. Multiple sections (5 mm) were prepared and stained with hematoxylin and eosin (H&E) for general morphological observation.

**Serum metabolomics**

*Metabolites extraction*. 50 μl of serum was used for metabolites extraction. 200 μl extract solution (acetonitrile:methanol = 1:1) containing isotopically labeled internal standard mixture was added to each sample. After vortexing for 1 min, the samples were sonicated for 10 min in cold water bath. Then, the samples were

incubated at −40 °C for 1 h, and centrifuged at $15,294 \times g$ for 15 min at 4 °C. 200 μl of supernatant were transferred to a fresh tube, and dried in a vacuum concentrator at 37 °C. Then, the dried samples were reconstituted in 200 μl of 50% acetonitrile by sonication on ice for 10 min. The constitution was then centrifuged at $17,949 \times g$ for 15 min at 4 °C, and 75 μl of supernatant were transferred to a fresh glass vial for LC/MS analysis.

*LC–MS/MS analysis*. The ultra-high-performance liquid chromatography (UHPLC) separation was carried out using a 1290 Infinity series UHPLC System (Agilent Technologies, USA), equipped with a UPLC BEH Amide column (2.1 * 100 mm, 1.7 μm, Waters, Milford, USA). The mobile phase consisted of 25 mmol/L ammonium acetate and 25 mmol/L ammonia hydroxide in water (pH = 9.75) (A) and acetonitrile (B). The analysis was carried with elution gradient as follows: 0–0.5 min, 95% B; 0.5–7.0 min, 95–65% B; 7.0–8.0 min, 65–40% B; 8.0–9.0 min, 40% B; 9.0–9.1 min, 40–95% B; 9.1–12.0 min, 95% B. The column temperature was 25 °C. The auto-sampler temperature was 4 °C, and the injection volume was 2 μl.

The TripleTOF 5600 mass spectrometry (AB Sciex, USA) was used for its ability to acquire MS/MS spectra on an information-dependent basis (IDA) during an LC/MS experiment. In this mode, the acquisition software (Analyst TF 1.7, AB Sciex) continuously evaluates the full scan survey MS data as it collects and triggers the acquisition of MS/MS spectra depending on preselected criteria. In each cycle, the most intensive 12 precursor ions with intensity above 100 were chosen for MS/MS at collision energy (CE) of 30 eV. The cycle time was 0.56 s. ESI source conditions were set as following: Gas 1 as 60 psi, Gas 2 as 60 psi, Curtain gas as 35 psi, Source temperature as 600 °C, declustering potential as 60 V, ion spray voltage floating (ISVF) as 5000 or −4000 V in positive or negative modes, respectively.

*Data preprocessing and annotation*. The acquired MS data pretreatments including peak selection and grouping, retention time correction, second peak grouping, and isotopes and adducts annotation, were performed as described previously[66] with a few modifications. UHPLC-MS raw data files were converted into mzXML format using the "msconvert" program from ProteoWizard and then analyzed by the XCMS[67] and CAMERA toolbox[68] with R statistical software. The CentWave algorithm in XCMS was used for peak detection. The parameter "peak-width" was set as (5, 20) in units of seconds, referring to the minimum and maximum peak widths for peak detection. The parameter "snthresh" is set as 3 for sensitive peak detection. For multiple UHPLC–MS data files, an ordered bijective interpolated warping (OBI-Warp) algorithm in XCMS was used for peak alignment[69]. By using retention time and the $m/z$ data pairs as the identifiers for each ion, we obtained ion intensities of each peak and generated a three dimensional matrix containing arbitrarily assigned peak indices (retention time–$m/z$ pairs), ion intensities (variables) and sample names (observations). Minfrac and cut off are set as 0.5 and 0.3, respectively. In-house MS2 database was applied for metabolites identification.

**Measurement of methylglyoxal by LC–MS/MS**. The MG levels in the serum and feces samples were measured using the LC–MS/MS methods cited in a previous study[70]. For the serum samples, 10 μl of ice-cold trichloroacetic acid (TCA)-saline (T6399, Sigma) was added to 20 μl of serum, and vortexed to mix. Next, 20 μl of water and 5 μl of [$^{13}C_3$] MG (739685, Sigma) (400 nM) were added. For the fecal samples, 10 mg of feces was homogenized with 20 μl ice-cold TCA-saline and 80 μl of water. Then, 5 μl [$^{13}C_3$] MG (400 nM) was added to the homogenized samples, and vortexed to mix. Next, the samples were centrifuged ($10,000 \times g$, 4 °C) for 10 min, and 35 μl of the supernatant was transferred to 200-μl glass inserts for 2-ml HPLC sample vials. The calibration standard assay solutions were prepared from a stock MG solution (800 nM), and transferred to 2-ml HPLC sample vials in glass inserts. Then, both the test samples and the calibration standard solution were derivatized to 2-methylquinoxaline (2-MQ) (W511609, Sigma) by adding 3% sodium azide (S2002, Sigma) and 0.5 mM 1,2-diaminobenzene (P23938, Sigma) in HCl-DETAPAC (D6518, Sigma), and the samples were incubated for 4 h in the dark at room temperature.

HPLC analyses were performed on an Acquity UPLC system with a Quattro Premier XE tandem mass spectrometric detector. The column was a BEH C18, 1.7-μm particle size, 100 × 2.1 mm column fitted with a 5 × 2.1 mm pre-column at 30 °C (Waters, Milford, USA). The sample temperature was maintained at 4 °C in the autosampler. The mobile phase was 0.1% (vol/vol) trifluoroacetic acid (TFA, 302031, Sigma) in water with a linear gradient of 0–100% solvent B over 10 min. The flow rate was 0.2 ml/min. The column was then washed for 5 min with 100% solvent B and re-equilibrated for 15 min with 100% solvent A; the flow rate was increased to 0.4 ml/min for this stage.

Mass spectrometric detection 2MQ and [$^{13}C_3$] 2MQ were detected by electrospray positive-ion multiple reaction monitoring (MRM) with a retention time of 8.0 min. The injection volume was 50 μl. MRM mass transition, collision energy (eV) and cone voltage (V) were as follows: 2MQ−145.1 > 77.1, 24 and 24; 145.1 > 92.1, 20 and 24; and [$^{13}C_3$] 2MQ−148.1 > 77.1, 24 and 24; 148.1 > 92.1, 20 and 24. Other mass spectrometer settings were as follows: capillary voltage, 0.60 kV; extractor voltage, 2.00 V; source temperature, 120 °C; desolvation gas temperature, 350 °C; desolvation gas flow, 900 liters per hour; and cone gas flow, 146 liters per hour.

**Cell culture**. HIB1B brown pre-adipocytes (kindly provided to us by Dr. Ke Ma at Beckman research Institute, Duarte, CA, USA) were cultured in Dulbecco's modified Eagle's medium (DMEM; Thermo Fisher, Pittsburg, PA, USA) supplemented with 10% fetal bovine serum (Atlas Biologicals, Fort Collins, CO, USA) and 1% penicillin–streptomycin (Thermo Fisher). After reaching confluence, cell differentiation was induced using high-glucose DMEM containing 10% FBS, 20 nM insulin, 1 nM triiodothyronine (T3), 0.5 mM IBMX, and 1 μM dexamethasone for 2 days. Cells were re-fed every other day with DMEM containing 10% fetal bovine serum and insulin and T3 at the same concentrations listed above. From the 1st day of the differentiation process, cells were treated with various concentrations of methylglyoxal. The HT-29 cell line was obtained from ATCC (HTB-38™, ATCC) and cultured in McCoy's 5a medium supplemented with 10% fetal bovine serum and 1% penicillin–streptomycin.

**IEC isolation**. Mice were sacrificed after anesthesia, and the small intestine was immediately removed and flushed from both ends with sterile PBS. The small intestine was then opened longitudinally and washed thoroughly with cold PBS. The tissue was then incubated in 10 ml of PBS containing 30 mM EDTA and 1.5 mM dithiothreitol (D0632, Sigma) on ice for 20 min, washed in cold PBS, and incubated in 10 ml of PBS containing 30 mM EDTA at 37 °C on a shaker for 10 min. Then, the samples were shaken vigorously for 30 s and centrifuged at $1000 \times g$ for 5 min at 4 °C. Cell pellets were washed with PBS containing 10% FBS and collected for experiments.

**AMPK knockdown in cells**. HT-29 cells were transfected with siRNA targeting PRKAA1 (sc-45312, Santa Cruz Biotechnology) or a control siRNA (SIC001, Sigma) using Lipofectamine 3000 (11668-027, Invitrogen) according to the manufacturer's instructions. Cells were incubated for 24 h with a transfection mixture containing a final siRNA concentration of 100 pM and then supplemented with fresh medium.

**Western blot analysis**. 20 μg of cells or 40 μg of tissue lysates were subjected to electrophoresis on 10% acrylamide gels and transferred to PVDF membranes. The membranes were incubated for 1 h with blocking buffer (either TBS-T containing 5% [w/v] BSA or 5% skim milk). The membranes were then incubated with the indicated primary antibodies diluted in blocking buffer (1:1,000) for 12 h at 4 °C: anti-pAmpk-α Thr172 (Cell Signaling, #2535s), anti-Ampk-α1 (R&D, #AF3197), anti-Ampk-α (Cell Signaling, #2532), anti-Ampk-α2 (abcam, ab3760), anti-UCP1 (abcam, ab10983), anti-HSP90 (Cell Signaling, #4874), anti-Bcl2 (abcam, ab196495), anti-cleaved Caspase 3 (Cell Signaling, #9661), anti-GAPDH (Cell Signaling, #2118), anti-Reg3γ (abcam, ab198216), and anti-Reg3α (abcam, ab202057). The membranes were washed three times with TBS-T and incubated with the secondary HRP-conjugated antibodies mouse anti-rabbit IgG-HRP (Cell Signaling, #7074S) (diluted 1:2000 in 5% skim milk) at room temperature for 1 h. Finally, the membranes were washed in TBS-T three times for 10 min each, and the signal was detected using enhanced chemiluminescence reagent (Pierce, IL, USA). Protein levels were quantified by densitometry using Image J software.

**RT-PCR analysis**. Total RNA was extracted from tissues and cells using TRIzol reagent, according to the manufacturer's instructions. The purity and concentration of the total RNA were determined using a NanoDrop spectrophotometer (ND-1000, Thermo Fisher). 1 μg of total RNA was reverse transcribed using cDNA Synthesis SuperMix (Invitrogen, CA, USA). Quantitative real-time PCR was carried out using an ABI 7900HT Fast Real-Time PCR System (Applied Biosystems, Warrington, UK) with SYBR Green PCR Master Mix (Applied Biosystems) and gene-specific primers (Supplementary Table 2). The sequences and GenBank accession numbers of the forward and reverse primers used to quantify mRNA are listed in Supplementary Table S2. Relative mRNA levels were normalized against Gapdh.

**Statistics and reproducibility**. Statistical analyses were performed using Graph-Pad Prism (version 8). All data are expressed as mean ± s.e.m. One-way and two-way analysis of variance (ANOVA) with Tukey's post-hoc tests were used to determine the statistical significance of differences between multiple groups. Student's $t$-tests were used to determine the statistical significance of differences between two groups. $P < 0.05$ was considered statistically significant. For the histology analysis and western blot analysis, each experiment was repeated three times independently with similar results.

**Reporting summary**. Further information on research design is available in the Nature Research Reporting Summary linked to this article.

## Data availability

The 16S rRNA sequencing data that support the findings of this study are available through the NCBI accession code PRJNA753804. The metabolomics data generated in this study are available through Metabolomics Workbench accession code ST001959 (https://doi.org/10.21228/M8NM5W) Source data are provided with this paper.

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

## Acknowledgements

We thank Dr. Fouad Kandeel, Dr. Debbie Thurmond, Dr. Sarah Highlander, and Dr. Qiong (Annabel) Wang for their helpful discussions. We particularly thank Dr. Kerin Higa for reviewing and editing the manuscript. This study was supported in part by the George & Irina Schaeffer Foundation, the John C. Hench Foundation, COH-UCR Biomedical Research Initiative, AR-DMRI Pilot Grant, Caltech-COH Biomedical Research Initiative, the National Institutes of Health (grants R01CA139158 and R01DK124627 to W.H. and COH P30CA33572), the National Natural Science Foundation of China (grants 81573581 to L.Y. and 8192010803 to Z.W.).

## Author contributions

W.H. conceptualized and designed the experiments, wrote and revised the manuscript. E.Z., L.Y. and L.J. designed the experiments, analyzed data, wrote and revised the manuscript. E.Z. performed most of the experiments. Y.W., J.T., L.D., Z.F., R.Z., and M.F. helped obtain the in vivo data. K.M., I.A., Z.W., S.C.S., R.N. and A.D.R. provided conceptual input and edited the manuscript.

## Competing interests

The authors declare no competing interests.
