## [Peer Review File · Nature Communications]

Reviewers' Comments:

Reviewer #1:

Remarks to the Author:

The manuscript by Zhang and colleagues reports the effect of intestinal-specific deletion of AMPK α 1 in mice. The authors showed that intestinal AMPK α 1 controls BAT function by modulating gut microbiota and production of methylglyoxal. They have raised the intriguing idea that metformin-induced intestinal AMPK activation promotes BAT thermogenesis. The strength of the study is that it is novel and provides potential link between intestinal AMPK and BAT function. The weakness of the study is that there is a tendency to overstate the claims, some critical experiments are missing to support the conclusions, and some of the data is difficult to interpret. Moreover, enthusiasm for the study is limited by the low n-numbers for some experiments. Lastly, I should add that the Discussion here is missing, and the reviewer think the authors should raise a number of key issues as listed below.

- In some cases the low n-numbers and/or a lack of quantification make it difficult to judge the relevance of the data. For instance, most of critical data (including gut microbiota composition, BAT gene expression profile and metabolomic analysis in Figure 2) are from n=3 or 4. Additionally, in Figure 1a, 2c, 2l, 4i, 4c and 4g there is no quantification of the lipid droplet accumulation in BAT or liver. Only single representative images are shown in these figures. Also, in most cases blots are shown with no quantification.
- One major question that arises in this reviewers mind is related to previous report showing that intestinal epithelium-specific deletion of AMPK α 1 protects mice from HFD-induced obesity and enhances glucose tolerance as well as insulin sensitivity (FASEB J. 2020 Apr;34(4):4852-4869. doi: 10.1096/fj.201901994R. Epub 2020 Feb 11. PMID: 32048347).
- The catalytic subunit of AMPK is expressed as two isoforms α 1 and α 2. Both have been reported to be expressed in the intestine. What about the expression of AMPK α 2 in AMPK α 1-IKO mice? Could the authors discuss on potential differential role for AMPK α 1 and α 2-containing AMPK complexes?
- An issue that the authors have not fully addressed is the mechanism(s) by which AMPK deletion in intestinal epithelial cells leads to the modulation of gut microbiota.
- There is also an important issue with the role of methylglyoxal (MG) in the system described here. From experiments conducted in DIO AMPK-IKO and AMPK α 1/fl mice, the authors conclude that intestinal AMPK deficiency modulates gut microbiota leading to higher MG levels which impairs BAT function (as demonstrated by decreased UCP1 expression, adipocyte hypertrophy associated with abundant lipid droplets, altered thermogenesis). However, it is not clear whether BAT dysfunction observed in chow diet-fed AMPK-IKO (Figure 1) is also relying on changes in methylglyoxal levels. Intriguingly, several recent investigations have suggested that MG is involved in many diseases such as diabetes and it is now recognized as a trigger for the development and progression of diabetic complications. Thus, how the authors can explain the absence of metabolic phenotype in chow diet-fed AMPK-IKO mice if methylglyoxal plays a key role?
- While almost obvious, the data as presented have not shown that the beneficial effects of metformin are due to a shift in gut microbiota and the consequent changes in circulating methylglyoxal levels. Is there any evidence that the effects of metformin in BAT is linked to changes in methylglyoxal levels? In addition, there is no demonstration that metformin treatment, besides reduction of lipid accumulation in BAT and changes in gut microbiota, is promoting a thermogenic program in DIO mice? What about oxygen consumption, heat production or response to cold exposure in metformin-treated mice?
- Could impaired thermogenesis in AMPK-IKO be rescued by using microbiota from WT or metformin-treated mice?

- Data have shown that germ-free mice have impaired thermogenesis of BAT, which is associated with limited increases in UCP1 expression and reduced browning of white adipose tissue (WAT). Is there any evidence that intestinal AMPK is required for the browning of WAT?
- To further document impaired thermogenesis in AMPK-IKO mice, some description of the effect of beta3-AR agonist on oxygen consumption would seem appropriate and necessary here.
- Following on from the previous comment, it is unclear how reduction of energy expenditure in AMPK-IKO mice on normal chow diet does not lead to increased bodyweight compared to WT.
- In figure 3J, it would be informative to show the p-AMPK/AMPK ratio to reflect AMPK activity. This is problematic because there seems to be no change in AMPK activity between non-T2D and T2D samples. The reviewers strongly feel that the authors need to amend their tone and downplay their interpretation.

Reviewer #2:

Remarks to the Author:

Mammalian AMPK is essential for glucose control and is a known therapeutic target for type 2 diabetes. However, the functions of AMPK in the intestine are mostly unknown. In this study, the authors assessed the phenotypes of intestinal epithelium-specific AMPK-null (AMPK-IKO) mice and discovered that intestinal AMPK is important for brown adipose tissue (BAT) thermogenesis. They showed a link between intestinal AMPK, anti-microbial peptide (AMP), the gut microbiome, and BAT. The authors suggested that this mechanism might partially underlie the therapeutic effects of metformin, which is an AMPK activator.

Overall, this is an interesting paper showing the role of intestinal AMPK in metabolism, including BAT thermogenesis, and a novel mechanistic link involving anti-microbial peptides and the gut microbiome and its metabolites. However, some of the conclusions are not supported by the data. Technical and biological replicates should be explicitly indicated to strengthen the experimental rigor. The paper will also benefit from having more details and a more extended discussion relating this study to previous literature. More specific comments are detailed below.

1. Figure 1: The manuscript only shows and describes BAT phenotype, but not other tissues. It would be critical to describe the changes in other tissues (white adipose tissues, muscle, liver, etc.) to help to interpret the overall phenotypes.

Lines 56-57 mention that food intake on HFD is similar between the knockout and the control mice, but food intake data is not shown. Please show these data. Also, for the other experiments throughout the paper, if food intake data were collected, it would be good to include them in the paper.

2. Figure 2:

- The number of mice is relatively low for the experiments in Figure 2 (n=3-4). Please indicate if these studies have been repeated in other cohorts. These experiments should be repeated to increase the number of mice. This is also the case for Supplementary Figures 2 and 3 and parts of Supplementary Figure 1 with n=4.
- Since mice are coprophagic, it is also important to note whether these mice were caged together or not and whether they came from the same litter.
- The method for Figure 2b is not explained anywhere. This should be described either in the figure legend or the Methods section.
- For the FMT experiment, how many mice were used as donors, and how were the donor mice selected? Were feces from all the donor mice mixed together, and all the recipient mice received the same fecal slurry? Or did each recipient mouse receive feces from a different donor?
- In the figure legend for Figure 2k, please provide a more detailed description of the methylglyoxal injection (e.g., dose and frequency of injection).
- Could the authors discuss more about the taxonomic composition of the gut microbiome? These data are shown in Supplementary Figure 2 but there is a lack of description about these taxa and how these findings relate to previous literature.
- For Figures 2c-f, 2i, and Supplementary Figure 3, how long after FMT these data were taken (how many days or weeks post-FMT)? Were the recipient mice fed with chow diet? These details

are missing in the manuscript.

3. The conclusions stated in lines 94-95 and lines 102-103 are not supported by the data. Here, the authors mention that increased methylglyoxal levels in AMPK-IKO mice are due to increased production of this metabolite by the gut microbiota. However, methylglyoxal is also produced by the host, as mentioned in lines 89-90. There is no evidence in the manuscript that the gut microbiome produces methylglyoxal in the AMPK-IKO mice. Intestinal AMPK knockout may lead to increased methylglyoxal production by the host. It is crucial to establish the origin of these metabolites highlighted in Figure 2g since one of the central claims in the paper is that gut microbiota metabolites mediate crosstalk between intestinal AMPK and BAT thermogenic regulation.

4. Figure 3: The conclusion (lines 123-125) is not strongly supported by the data. This statement says that AMPK modulates the gut microbiome through Reg3 γ . However, the results only show that intestinal AMPK regulates Reg3, but no data is showing that Reg3 regulates the gut microbiome or their metabolites (e.g. methylglyoxal). There may be no causal link between Reg3 expression in these mice and the gut microbiome.

5. Figure 4: These results are quite striking, and it would be helpful to describe these in the context of previous findings. Other studies have shown the importance of liver AMPK in mediating metformin's actions. The authors should provide a discussion comparing the results of this study with these previously published results. The discussion may include the extent to which intestinal AMPK, relative to AMPK in other tissues, might mediate metformin's effects.

- The authors could elaborate more on some of the details and relate the microbiome findings with previous literature. Are the metformin-induced microbiome alterations shown in this study similar to those shown in previous studies? If the metformin-treated mice's microbiome profiles are different from previous findings, what are some possible explanations?

6. In general, bar charts would be clearer if represented as box plots with individual data points shown, so that the distribution of the data can be seen. Whenever possible, quantify the western blots and histology images.

The manuscript has a few grammatical errors and typos, including:

- For Supplementary Figure 2 legend, the last sentence is missing a period before "Values".
- Line 79 is missing "that" before "received FMT".
- Figure 2f legend: remove "mice" before "recipient mice".
- Line 87: "funcions" should be "functions" instead.
- Supplementary Figure 1: The figure legend describes figures s-t, but there is no figure s or t.

Reviewer #3:

Remarks to the Author:

Zhang et al. reported 'intestinal AMPK modulation of microbiota mediates cross-talk with brown fat to control thermogenesis'. This study is timely, well-written, and well-executed.

The authors genetically deleted AMPK alpha 1 in the IECs. AMPK-IKO mice on chow diet had marked adipocyte hypertrophy and increased lipid droplets as well as down-regulation of thermogenic gene program, and altered genes in mitochondrial function, fatty acid oxidation, and lipolytic pathways, together with an inhibition on energy expenditure. On a HFD, AMPK-IKO mice gained more weight, developed impaired glucose and insulin tolerance. Further, changes in gut microbiome were also detected in AMPK-IKO mice on chow diet and FMT of the gut microbiome obtained from AMPK-IKO replicated the metabolic phenotypes in AMPK-IKO mice. Next, the authors discovered that AMPK-IKO mice had higher levels of microbiota-derived methylglyoxal and the effect was replicated in FMT of the gut microbiome from AMPK-IKO mice. More importantly, direct administration of methylglyoxal per se replicated the phenotypes of AMPK-IKO mice. The Authors then reported AMPK-IKO mice had reduced intestinal Reg3 γ , and that metformin increased intestinal Reg3 γ expression in control but not AMPK-IKO mice. Interestingly, people with obesity and T2D also had reduced duodenal AMPK activity as well as Reg3 γ expression. Then the authors

found that metformin exerted benefits (i.e., weight loss and improvement on glucose tolerance) as well as changes in gut microbiome were attenuated in AMPK-IKO mice.

I only have one question: Re the glucose tolerance observation reported in Figure 1h and Figure 4b, is the effect dependent on weight changes? Would the authors still observe the same trend if weight changes were controlled for?

Reviewer #4:

Remarks to the Author:

The goal of this manuscript by Zhang et al is to demonstrate that intestinal AMPK and brown adipose tissue communicate via gut microbiota-derived metabolites. Overall, this is a robust manuscript that reports interesting communication between intestine and brown adipose tissue. Nevertheless, I have the following significant concerns with regards to metabolomics experimentation and metabolomics data interpretation:

1. Methods leading to Figure 2b are missing.
2. Metabolomics methods are insufficient. Missing information includes: metabolite extraction method, all LC-MS parameters and all data analysis parameters. Authors are referred to the Metabolomics Standards Initiative (PMC3772505) for a comprehensive list of the information that needs to be provided.
3. Authors must provide evidence to support the metabolite annotations provided in Figure 2g. Acceptable evidence would be MS2 spectral matching to reference spectrum, or matched extracted ion chromatograms compared to a pure standard. This is particularly important given multiple unlikely metabolite annotations in Figure 2g, such as prilocaine, gaboxadol, coumarin, exalamide, perillyl alcohol, pargyline and lavandulol.
4. Given these dubious annotations, confirming methylglyoxal annotation to the standards described above is essential, particularly since it is the basis for multiple subsequent experiments.
5. I would recommend that the authors consider publicly depositing their metabolomics data, as is standard for sequencing data.
6. A central tenet of this manuscript is that intestine-AMPK communication is via microbiota metabolites, specifically methylglyoxal. However, this metabolite is ubiquitous and can also be produced by human cells. Elevated methylglyoxal in cecum contents rather than serum would be a more convincing indication of microbiota source than serum analyses, as would indication that it is produced by bacteria cultured from IKO microbiota.
7. Likewise, confirming that increased serum methylglyoxal is a cause of the adipose tissue phenotype requires confirmation of elevated methylglyoxal in BAT. Otherwise, altered BAT phenotype could be an indirect effect.
8. Methylglyoxal is toxic. Authors should provide additional experiments to confirm that the effects of methylglyoxal on UCP1 levels are not merely a consequence of cell death.
9. How do the authors explain the effects of methylglyoxal, given endogenous mechanisms for detoxifying it (e.g. glyoxalase), and the fact that methylglyoxal is rapidly bound to macromolecules?
10. Authors only describe intestinal epithelial cell isolation from the small intestine (lines 265-271). Given this manuscript's emphasis on AMPK-microbiota interaction, it is essential to confirm loss of AMPK in the KO mice in the cecum and large intestine as well as the small intestine.
11. Though supplementary figure S1m and S1n data is marked as statistically significance, I question the biological validity of these findings, given the small magnitude of change.

Minor comments:

1. Figure 2j: what is the EC50 of methylglyoxal with regards to UCP1 expression in HIB1B cells?
2. Lines 138-139: "We found that metformin reduced lipid accumulation and UCP1 level in the BAT of AMPKfl/fl, but not AMPK-IKO mice". In fact, metformin appears to increase UCP1 in control mice (Figure 4j). Is this a typo/misphrasing?
3. Densitometry should be provided for Figure 4j, where differences between groups are harder to see (especially in IKO mice).
4. Lines 99-100: "Furthermore, we show that oral gavage of mice with methylglyoxal reduced UCP1 expression in BAT" but methods indicate "mice were treated with intraperitoneal (i.p.) injections of methylglyoxal (50 mg/kg) once daily for 14 days." (lines 202-203). Which route was

actually used?

Point-by-point response to the reviewers' comments

Reviewer #1 (Remarks to the Author):

The manuscript by Zhang and colleagues reports the effect of intestinal-specific deletion of AMPK α 1 in mice. The authors showed that intestinal AMPK α 1 controls BAT function by modulating gut microbiota and production of methylglyoxal. They have raised the intriguing idea that metformin-induced intestinal AMPK activation promotes BAT thermogenesis. The strength of the study is that it is novel and provides potential link between intestinal AMPK and BAT function. The weakness of the study is that there is a tendency to overstate the claims, some critical experiments are missing to support the conclusions, and some of the data is difficult to interpret. Moreover, enthusiasm for the study is limited by the low n-numbers for some experiments. Lastly, I should add that the Discussion here is missing, and the reviewer think the authors should raise a number of key issues as listed below.

We appreciate the positive and insightful comments from this reviewer. Please find below our point-by-point response to individual comment.

1. - In some cases the low n-numbers and/or a lack of quantification make it difficult to judge the relevance of the data. For instance, most of critical data (including gut microbiota composition, BAT gene expression profile and metabolomic analysis in Figure 2) are from n=3 or 4. Additionally, in Figure 1a, 2c, 2l, 4i, 4c and 4g there is no quantification of the lipid droplet accumulation in BAT or liver. Only

Response Letter Figure 1. The quantification of adipocyte size and western blots bands. (a) The quantification of adipocyte size in manuscript Fig. 1a. (b) The quantification of western blots bands in manuscript Fig. 1c. (c) The quantification of western blots bands in manuscript Fig. 1m. (d) The quantification of adipocyte size in manuscript Fig. 2c. (e) The quantification of western blots bands in manuscript Fig. 2e. (f) The quantification of western blots bands in manuscript Fig. 2k. (g) The quantification of adipocyte size in manuscript Fig. 2l. (h) The quantification of western blots bands in manuscript Fig. 3b. (i) The quantification of western blots bands in manuscript Fig. 3h. (j) The quantification of lipid droplet size in manuscript Fig. 4c and 4h. (k) The quantification of adipocyte size in manuscript Fig. 4k. (l) The quantification of western blots bands in manuscript Fig. 4l. * $p < 0.05$, ** $p < 0.01$ and *** $p < 0.001$ by two-tailed Student's t-tests or two-way ANOVA with Tukey's post-hoc tests

single representative images are shown in these figures. Also, in most cases blots are shown with no quantification.

For the Figure 2 and supplementary Figures 1-3 with n=3 or 4, we repeated in other cohorts of mice and updated these figures with final results in the revised manuscript.

As the reviewer requested, we also quantified the lipid droplet accumulation in BAT and liver in all the HE stained sections and added these results into the manuscript as new Supplementary Figure 1e, 3a, 4c, 6c and 6e, respectively. In addition, we quantified the western blots and added the results in the manuscript as new Supplementary Figure 1f, 1r, 3b, 4b, 5b, 5d and 6f, respectively, as shown in **Response Letter Fig. 1**.

2. - One major question that arises in this reviewers mind is related to previous report showing that intestinal epithelium-specific deletion of *AMPKalpha1* protects mice from HFD-induced obesity and enhances glucose tolerance as well as insulin sensitivity (FASEB J. 2020 Apr;34(4):4852-4869. doi: 10.1096/fj.201901994R. Epub 2020 Feb 11. PMID: 32048347).

Regarding the discrepant phenotype, the different genetic backgrounds might be one of the reasons. In our study, we used the Villin-Cre mice (B6. Cg-Tg (Vil-cre) 1000Gum/J, stock No. 021504) purchased from Jackson Lab. In the report of PMID 32048347, they used the Villin-Cre mice (B6.Cg-Tg(Vil1-cre)997Gum/J, Stock No. 004586) from Jackson Lab. It has been reported that the mice of stock No. 021504 are absent of Cre recombinase activity in gonads; while the mice of stock No. 004586 show Cre recombinase activity in a very low level (<1%) of cells in the testes (<https://www.jax.org/strain/021504>).

In the report of PMID 32048347, to explain their observations on the phenotype, the authors provide a mechanism that AMPK facilitates intestinal long-chain fatty acid uptake by manipulating CD36 expression and translocation. However, several studies suggest that the regulation of CD36 expression is AMPK independent (Jeppesen et al. 2011, Pinkosky et al. 2020). Pinkosky et al demonstrated that AMPK is not essential in regulation of FAT/CD36 translocation and FA uptake in skeletal muscle during contractions. We also examined the gene expression of CD36 and FATP4 in the small intestine of our cohorts of mice and didn't observe significant changes (**Response Letter Fig. 2**).

On the other hand, it has been reported that compared with normal controls, diabetic gerbils possessed lower level of AMPK phosphorylation in the small intestine, contributing to the impaired metabolism by regulating pathways of insulin signaling, FA oxidation, and lipid/lipoprotein synthesis (Harmel et al. 2014). In our study, we observed significantly reduced AMPK phosphorylation in the duodenum samples from obese patients with T2D, further confirming the lower activation of intestinal AMPK in T2D condition. These results thus support that intestinal AMPK is required in maintaining the homeostasis of glucose and lipid metabolism.

References:

Harmel E, et al. AMPK in the Small Intestine in Normal and Pathophysiological Conditions. *Endocrinology* **155**, 873-888 (2014).

Jeppesen J, et al. Contraction-induced skeletal muscle FAT/CD36 trafficking and FA uptake is AMPK independent. *Journal of lipid research* **52**, 699-711 (2011).

Response Letter Figure 2. CD36 and FATP4 gene expression in the jejunum tissue of AMPK^{fl/fl} (Control) and AMPK-IKO mice fed ND chow.

Pinkosky SL, et al. Long-chain fatty acyl-CoA esters regulate metabolism via allosteric control of AMPK β 1 isoforms. *Nature metabolism* **2**, 873-881 (2020).

3. - The catalytic subunit of AMPK is expressed as two isoforms α 1 and α 2. Both have been reported to be expressed in the intestine. What about the expression of AMPK α 2 in AMPK α 1-IKO mice? Could the authors discuss on potential differential role for AMPK α 1 and α 2-containing AMPK complexes?

We detected the expression of AMPK α 2 in IECs isolated from our AMPK α 1-IKO mice, and we didn't observe significant change (Response Letter Fig. 3).

Response Letter Figure 3. AMPK α 2 protein expression in the IEC of AMPK α 1-IKO (Control) and AMPK-IKO mice fed ND chow.

AMPK catalytic subunit includes two isoforms (α 1 and α 2) that are enriched in specific tissues. AMPK α 1 is enriched in spleen and brain, while AMPK α 2 is enriched in liver, skeletal muscle, kidney, and heart (Stapleton et al. 1996; Lihn et al. 2004; Cheung et al. 2000). Furthermore, the 2 isoforms show different cell locations. AMPK α 1 subunit is distributed mainly in the cytosol, whereas the α 2 is also located in the nucleus (Salt et al. 1998). The different tissue distribution and cell location of AMPK α 1 and α 2 suggest a specialized role of AMPK α 1 and α 2-containing AMPK complexes in functional regulation. For example, expression of a constitutively active AMPK α 2 in the liver reduced blood glucose and gluconeogenic genes in wild-type and ob/ob diabetic mice (Foretz et al. 2005), while the mice with a liver-specific knockout of AMPK α 2 were hyperglycemic and had elevated HGP relative to controls (Andreelli et al. 2006). However, mice lacking both AMPK α 1 and α 2 subunits in the liver did not result in hyperglycemia or increases in HGP. Moreover, in human skeletal muscle, exercise activates the AMPK complex containing the α 2/ β 2/ γ 3 heterotrimer whereas the activities of complexes containing the α 1 catalytic subunit remain unchanged (Birk et al. 2006).

Recent studies reported that mice harboring an intestinal-specific ablation of AMPK α 1 display impairment in intestinal barrier function, epithelial differentiation, and long-chain fatty acid uptake (Sun et al. 2017; Wu et al. 2020). When the AMPK α 1 and AMPK α 2 were both specifically knockout in the intestine, the mice showed leaky gut barrier in the distal colon but not change in the body weight gain and glucose homeostasis (Olivier et al. 2021). It has been reported that in the intestine, AMPK is mainly expressed in IECs and the predominance of AMPK α subunit is AMPK α 1 (Elodie Harmel et al. 2014, Olivier et al. 2021), thus we selected intestinal AMPK α 1 to investigate its role in energy and metabolism in our current study, and uncovered a novel link between intestinal AMPK activation and BAT thermogenic regulation through modulating anti-microbial peptide (AMP)-controlled gut microbiota and the metabolites.

References:

- Andreelli F, et al. Liver adenosine monophosphate-activated kinase- α 2 catalytic subunit is a key target for the control of hepatic glucose production by adiponectin and leptin but not insulin. *Endocrinology* **147**, 2432-2441 (2006).
- Birk JB, Wojtaszewski JF. Predominant α 2/ β 2/ γ 3 AMPK activation during exercise in human skeletal muscle. *The Journal of physiology* **577**, 1021-1032 (2006).
- Cheung PC, et al. Characterization of AMP-activated protein kinase γ -subunit isoforms and their role in AMP binding. *The Biochemical journal* **346 Pt 3**, 659-669 (2000).
- Foretz M, et al. Short-term overexpression of a constitutively active form of AMP-activated protein kinase in the liver leads to mild hypoglycemia and fatty liver. *Diabetes* **54**, 1331-1339 (2005).
- Harmel E, et al. AMPK in the Small Intestine in Normal and Pathophysiological Conditions. *Endocrinology* **155**, 873-888 (2014).

Lihn AS, et al. AICAR stimulates adiponectin and inhibits cytokines in adipose tissue. *Biochem Biophys Res Commun* **316**, 853-858 (2004).

Olivier S, et al. Deletion of intestinal epithelial AMP-activated protein kinase alters distal colon permeability but not glucose homeostasis. *Mol Metab.* **47**, 101183 (2021).

Salt I, et al. AMP-activated protein kinase: greater AMP dependence, and preferential nuclear localization, of complexes containing the alpha2 isoform. *The Biochemical journal* **334** (Pt 1), 177-187 (1998).

Stapleton D, et al. Mammalian AMP-activated protein kinase subfamily. *The Journal of biological chemistry* **271**, 611-614 (1996).

Sun X, et al. AMPK improves gut epithelial differentiation and barrier function via regulating Cdx2 expression. *Cell death and differentiation* **24**, 819-831 (2017).

Ross FA, et al. AMP-activated protein kinase: a cellular energy sensor that comes in 12 flavours. *FEBS J* **283**, 2987-3001 (2016).

Wu W, et al. AMPK facilitates intestinal long-chain fatty acid uptake by manipulating CD36 expression and translocation. *FASEB journal: official publication of the Federation of American Societies for Experimental Biology* **34**, 4852-4869 (2020).

4. - An issue that the authors have not fully addressed is the mechanism(s) by which AMPK deletion in intestinal epithelial cells leads to the modulation of gut microbiota.

To investigate the mechanisms underlying GM alterations in AMPK-IKO mice, we measured the expressions of several epithelium-associated factors known to play roles in modulating GM, including AMPs, innate immune sensors, the mucus barrier, secretory IgAs, epithelial microvilli, and epithelial tight junctions (Chang et al. 2019). Among the tested genes, AMPs including Reg3 β and Reg3 γ were significantly decreased in the jejunum, duodenum, ileum and colon tissues of AMPK IKO mice (**Response Letter Fig. 4**). In addition, significantly decreased expression of RELM β and MMP7 genes were also observed in the jejunum and duodenum tissues in AMPK IKO mice.

Our original manuscript has demonstrated the dependent role of intestinal AMPK on AMPs levels (**manuscript Fig. 3** and **Supplementary Fig. 5**). Together with numerous evidence indicating that AMPs are epithelial innate antibiotic effector molecules with important roles in regulating the gut microbial community and ensuring a beneficial homeostasis at the intestinal barrier (Ostaff et al. 2013; Natividad et al. 2013; Loonen et al. 2014; Cash et al. 2006; Morampudi et al. 2016), our results suggest that intestinal AMPK may modulate GM profile through regulating the levels of AMPs, providing a potential mechanism by which AMPK deletion in intestinal epithelial cells leads to the modulation of gut microbiota. The exact molecular mechanism by which intestinal AMPK regulates AMP expression in the intestinal epithelial cells will be our future endeavor.

References:

Cash HL, et al. Symbiotic Bacteria Direct Expression of an Intestinal Bactericidal Lectin. *Science (New York, NY)* **313**, 1126-1130 (2006).

Chang C-S, et al. Current understanding of the gut microbiota shaping mechanisms. *Journal of Biomedical Science* **26**, 59 (2019).

Loonen LM, et al. REG3 γ -deficient mice have altered mucus distribution and increased mucosal inflammatory responses to the microbiota and enteric pathogens in the ileum. *Mucosal Immunol* **7**, 939-947 (2014).

Morampudi V, et al. The goblet cell-derived mediator RELM- β drives spontaneous colitis in Muc2-deficient mice by promoting commensal microbial dysbiosis. *Mucosal immunology* **9**, 1218-1233 (2016).

Natividad JM, et al. Differential induction of antimicrobial REGIII by the intestinal microbiota and *Bifidobacterium breve* NCC2950. *Applied and environmental microbiology* **79**, 7745-7754 (2013).

Ostaff MJ, et al. Antimicrobial peptides and gut microbiota in homeostasis and pathology. *EMBO molecular medicine* 5, 1465-1483 (2013).

Response Letter Figure 4. Epithelium-associated factors gene expression in jejunum, duodenum, ileum and colon tissues in the AMPK^{fl/fl} (Control) and AMPK-IKO mice fed ND chow. n=4-6/group. Values are means ± s.e.m. *p<0.05, ** p<0.01 and ***p<0.001 by two-tailed Student's t-tests.

5. - There is also an important issue with the role of methylglyoxal (MG) in the system described here. From experiments conducted in DIO AMPK-IKO and AMPK^{fl/fl} mice, the authors conclude that intestinal AMPK deficiency modulates gut microbiota leading to higher MG levels which impairs BAT function (as demonstrated by decreased UCP1 expression, adipocyte hypertrophy associated with abundant lipid droplets, altered thermogenesis). However, it is not clear whether BAT dysfunction observed in chow diet-fed AMPK-IKO (Figure 1) is also relying on changes in methylglyoxal levels.

Intriguingly, several recent investigations have suggested that MG is involved in many diseases such as diabetes and it is now recognized as a trigger for the development and progression of diabetic complications. Thus, how the authors can explain the absence of metabolic phenotype in chow diet-fed AMPK-IKO mice if methylglyoxal plays a key role?

We appreciate the reviewer's insightful comments. We also conducted the serum metabolomics analysis in the serum of chow diet-fed AMPK IKO mice using the UHPLC-QTOF-MS method. As expected, serum MG levels were higher in AMPK IKO mice than WT mice (**Response Letter Fig.5**). The lower energy expenditure and impaired BAT thermogenesis in chow diet-fed AMPK IKO mice, together with the result that MG administration reduced UCP1 expression and induced lipid accumulation in BAT (**manuscript Fig. 2k-l, Supplementary Fig. 4a-c**) suggest a potential role of MG in BAT function and energy expenditure.

Although lots of evidence have shown increased MG levels correlated with hyperglycemia and insulin resistance, whether MG can trigger chow diet-fed mice to diabetes remains uncertain. For example, it was reported that MG-supplemented rats fed with chow diet displayed structural and functional alterations in adipose tissue, but did not develop insulin resistance or hyperglycemia (Matafome et al. 2012; Rodrigues et al. 2013).

References:

Matafome P, et al. Methylglyoxal causes structural and functional alterations in adipose tissue independently of obesity. *Archives of physiology and biochemistry* **118**, 58-68 (2012).

Rodrigues T, et al. Reduction of methylglyoxal-induced glycation by pyridoxamine improves adipose tissue microvascular lesions. *Journal of diabetes research* **2013**, 690650 (2013).

6. - While almost obvious, the data as presented have not shown that the beneficial effects of metformin are due to a shift in gut microbiota and the consequent changes in circulating methylglyoxal levels. Is there any evidence that the effects of metformin in BAT is linked to changes in methylglyoxal levels? In addition, there is no demonstration that metformin treatment, besides reduction of lipid accumulation in BAT and changes in gut microbiota, is promoting a thermogenic program in DIO mice? What about oxygen consumption, heat production or response to cold exposure in metformin-treated mice?

Thank you for this insightful comment. As suggested, we measured the methylglyoxal levels in metformin treated mice, and found that the serum methylglyoxal levels in the DIO mice were reduced by metformin in AMPK^{fl/fl} mice but not AMPK IKO mice (**Response Letter Fig. 6**).

Together with the results in **manuscript Fig. 2j-l**, our results suggest that the effects of metformin in BAT is linked to changes in methylglyoxal levels. Previous reports also support this potential link (Beisswenger et al. 1999; Kender et al. 2014).

Response Letter Figure 5. Metabolomic analysis of serum samples from AMPK^{fl/fl} (Control) and IKO mice fed ND chow. *p<0.05 by two-tailed Student's t-tests.

Response Letter Figure 6. Serum Methylglyoxal levels in the metformin-treated DIO-AMPK control mice (a) and DIO-AMPK IKO mice (b). Mice were orally gavage with metformin (100 mg/kg) once daily for 8 weeks. ** p<0.01 by two-tailed Student's t-tests.

As suggested, we also examined the effect of metformin on the energy expenditure by indirect calorimetry. We observed that metformin-treated mice consumed significantly more O₂, exhaled more CO₂, indicating the increased energy expenditure after metformin treatment (**Response Letter Fig. 7**), which is consistent with a previous study (Breining et al. 2018). We also performed a cold tolerance test for these mice and didn't observe significant change in heat production between metformin- and vehicle-treated mice. These results indicated that metformin treatment is promoting energy expenditure in DIO mice.

References:

Response Letter Figure 7. VO₂, VCO₂ and energy expenditure in metformin-treated DIO WT mice. Mice were orally gavage with metformin (100 mg/kg) or vehicle control (PBS) once daily for 8 weeks. Values are means ± s.e.m. *p<0.05 by two-tailed Student's t-tests.

Beisswenger PJ, et al. Metformin reduces systemic methylglyoxal levels in type 2 diabetes. *Diabetes* **48**, 198-202 (1999).

Kender Z, et al. Effect of metformin on methylglyoxal metabolism in patients with type 2 diabetes. *Experimental and clinical endocrinology & diabetes : official journal, German Society of Endocrinology [and] German Diabetes Association* **122**, 316-319 (2014).

Breining P, et al. Metformin targets brown adipose tissue in vivo and reduces oxygen consumption in vitro. *Diabetes, obesity & metabolism* **20**, 2264-2273 (2018).

7.- Could impaired thermogenesis in AMPK-IKO be rescued by using microbiota from WT or metformin-treated mice?

To answer this question, we performed an FMT by transplanting the fecal microbiome from AMPK^{fl/fl} mice to the antibiotic-induced microbiome depletion (AIMD)- AMPK IKO mice, using fecal samples from AMPK-IKO mice as a control. The AMPK IKO mice received FMT from AMPK^{fl/fl} mice (FMT-CK) showed smaller adipocyte (**Response Letter Fig. 8 a-b, Supplementary Fig. 3h-i**) and higher UCP1 levels (**Response Letter Fig. 8 c-e, Supplementary Fig. 3j-k**) compared with the AMPK-IKO mice received FMT from AMPK IKO mice (FMT-KK). When the recipient mice were put into cold environment, the FMT-CK mice showed higher body temperature than the FMT-KK mice (**Response Letter Fig.8f**,

Supplementary Fig. 3m). These results demonstrated that the impaired thermogenesis in AMPK-IKO mice can be rescued by FMT from WT mice.

Response Letter Figure 8. The impaired thermogenesis in AMPK-IKO mice can be rescued by using microbiota from WT mice. FMT-CK, FMT from AMPK^{fl/fl} control mice to AMPK-IKO mice; FMT-KK, FMT from AMPK-IKO mice to AMPK-IKO mice. (a-b) Representative images and adipocyte size analysis of H&E-stained BAT sections from FMT recipient mice. Scale bar = 100 μm . (c-d) Representative Western blot analysis of UCP1 protein levels in the BAT of recipient mice. (e) Relative mRNA levels of genes expressed in the BAT of recipient mice. (f) Rectal temperatures of recipient mice exposed to 6°C for 2 h. * $p < 0.05$, ** $p < 0.01$ and *** $p < 0.001$ by two-tailed Student's t-tests (for b, d, e) or two-way ANOVA with Tukey's post-hoc tests for f.

8. - Data have shown that germ-free mice have impaired thermogenesis of BAT, which is associated with limited increases in UCP1 expression and reduced browning of white adipose tissue (WAT). Is there any evidence that intestinal AMPK is required for the browning of WAT?

Besides the phenotype in BAT, we also assessed phenotypes in WAT in the AMPK-IKO mice. We didn't observe significant change in the morphology of WAT adipocyte size, as well as the UCP1 protein levels in WAT of mice fed chow diet or HFD (**Response Letter Fig.9**), indicating that intestinal AMPK is not required for the browning of WAT.

Response Letter Figure 9. WAT phenotype in AMPK-IKO mice. (a-b) Representative images and analysis of H&E-stained inguinal WAT sections in chow diet and HFD-fed AMPK^{fl/fl} and IKO mice. scale bar = 100 μm . (c-d) Western blot analysis of UCP1 protein levels in the iWAT mice fed ND chow (c) and HFD (d).

9. - To further document impaired thermogenesis in AMPK-IKO mice, some description of the effect of beta3-AR agonist on oxygen consumption would seem appropriate and necessary here.

Thank you for the suggestions. One area of focus for over two decades has been the stimulation of BAT energy expenditure and nutrient consumption through activation of the β 3-adrenergic receptor (AR) (Virtanen et al. 2009; Cypess et al. 2015; Cypess et al. 2013; Ursino et al. 2009). Administration of CL-316,243, a potent and highly selective β 3-AR agonist (Bloom et al. 1992) leads to marked increases in thermogenesis by BAT (Susulic et al. 1995). In our study, we observed significantly lower β 3-AR gene level in the BAT of AMPK IKO mice compared with that of AMPK control mice (**manuscript Fig.1b**), suggesting that the impaired thermogenesis effect may be related to the reduced activity of β 3-AR.

References:

Bloom JD, et al. Disodium (R,R)-5-[2-[[2-(3-chlorophenyl)-2-hydroxyethyl]-amino] propyl]-1,3-benzodioxole-2,2-dicarboxylate (CL 316,243). A potent beta-adrenergic agonist virtually specific for beta 3 receptors. A promising antidiabetic and antiobesity agent. *Journal of medicinal chemistry* **35**, 3081-3084 (1992).

Cypess AM, et al. Activation of human brown adipose tissue by a β 3-adrenergic receptor agonist. *Cell metabolism* **21**, 33-38 (2015).

Cypess AM, et al. Anatomical localization, gene expression profiling and functional characterization of adult human neck brown fat. *Nature medicine* **19**, 635-639 (2013).

Susulic VS, et al. Targeted disruption of the beta 3-adrenergic receptor gene. *The Journal of biological chemistry* **270**, 29483-29492 (1995).

Ursino MG, et al. The beta3-adrenoceptor as a therapeutic target: current perspectives. *Pharmacological research* **59**, 221-234 (2009).

Virtanen KA, et al. Functional brown adipose tissue in healthy adults. *The New England journal of medicine* **360**, 1518-1525 (2009).

10. - Following on from the previous comment, it is unclear how reduction of energy expenditure in AMPK-IKO mice on normal chow diet does not lead to increased bodyweight compared to WT.

We observed lower energy expenditure in chow diet fed-AMPK IKO mice, which was consistent with the BAT dysfunction. The UCP1/BAT system has evolved to maintain body temperature (Kozak LP et al. 2010) as the primary role in body, whereas, leading to weight loss is secondary to the primary role (Kozak LP et al. 2010; Kozak LP et al. 2014). Moreover, mice deficient in UCP1 do not have increased susceptibility to obesity (N.J. Rothwell et al. 1979). Therefore, it may need more pressive stimulus such as HFD-feeding or cold challenge to induce the secondary response of this kind of gene deficient mice. That is consistent with our result that chow diet-fed AMPK IKO mice only exhibited lower energy expenditure while HFD-fed IKO mice showed more severely metabolic disorders including impaired energy expenditure as well as insulin resistance and weight gain (**manuscript Fig. 1 and Supplementary Fig. 1**).

References:

Kozak LP. Brown fat and the myth of diet-induced thermogenesis. *Cell Metab* **11**, 263-267 (2010).

Kozak LP. Genetic variation in brown fat activity and body weight regulation in mice: Lessons for human studies, *Biochimica et Biophysica Acta (BBA) - Molecular Basis of Disease* **1842**, 370-376 (2014).

Rothwell NJ, et al. A role for brown adipose tissue in diet-induced thermogenesis. *Nature* **281**,31-35. (1979).

11. - In figure 3J, it would informative to show the p-AMPK/AMPK ratio to reflect AMPK activity. This is problematic because there seems to be no change in AMPK activity between non-T2D and T2D

samples. The reviewers strongly feel that the authors need to amend their tone and downplay their interpretation.

Thank you for the insightful suggestion. As suggested, we calculated the ratio of p-AMPK/AMPK for the western blotting analysis in manuscript Figure 3j. The result (**Response Letter Figure 10**) showed no difference between Non-T2D and T2D samples. Because the patient mucosa samples include different types of cells such as immune cells, muscle cells and epithelial cells, AMPK protein levels in the mucosa samples could not exactly present the AMPK levels in the epithelial cells only. Our results showed that the levels of p-AMPK in the mucosa samples from T2D patients were significantly lower than that from control group, suggesting a potential role of intestinal AMPK in regulating glucose homeostasis and T2D pathogenesis. Accordingly, we have modified the description about this result in the revised manuscript.

Response Letter Figure 10.
The ratio of the quantified densities of p-AMPK/AMPK in the Western blots in manuscript Figure 2e.

Reviewer #2 (Remarks to the Author):

Mammalian AMPK is essential for glucose control and is a known therapeutic target for type 2 diabetes. However, the functions of AMPK in the intestine are mostly unknown. In this study, the authors assessed the phenotypes of intestinal epithelium-specific AMPK-null (AMPK-IKO) mice and discovered that intestinal AMPK is important for brown adipose tissue (BAT) thermogenesis. They showed a link between intestinal AMPK, anti-microbial peptide (AMP), the gut microbiome, and BAT. The authors suggested that this mechanism might partially underlie the therapeutic effects of metformin, which is an AMPK activator.

Overall, this is an interesting paper showing the role of intestinal AMPK in metabolism, including BAT thermogenesis, and a novel mechanistic link involving anti-microbial peptides and the gut microbiome and its metabolites. However, some of the conclusions are not supported by the data. Technical and biological replicates should be explicitly indicated to strengthen the experimental rigor. The paper will also benefit from having more details and a more extended discussion relating this study to previous literature. More specific comments are detailed below.

We appreciate the positive and insightful comments from this reviewer. Please find below our point-by-point response to individual comment.

1. Figure 1: The manuscript only shows and describes BAT phenotype, but not other tissues. It would be critical to describe the changes in other tissues (white adipose tissues, muscle, liver, etc.) to help to interpret the overall phenotypes.

We have examined phenotypes in various tissues of our newly generated AMPK IKO mice, including histological examination of muscle, liver, WAT, BAT and different sections of intestine, and functional gene expressions. Except for the BAT, we didn't observe significant change in the morphology of these tissues, as well as the expression of genes related with lipid and glucose metabolism, in mice fed normal chow diet (**Response Letter Figure 9, 11**). Therefore, in this study, we focused on BAT to investigate the role of intestinal AMPK in energy and metabolism.

For the liver tissue, we didn't observe any changes in the histological examination of liver sections from AMPK control mice and IKO mice (**Response Letter Fig.11g**). However, the genes involved in the gluconeogenesis were significantly increased in the AMPK IKO mice compared to AMPK control mice

Response Letter Figure 11. Examinations were performed in the HFD-fed AMPK^{fl/fl} and IKO mice. (a-b) Representative images and adipocyte size analysis of H&E-stained inguinal WAT (iWAT) sections. (c) Representative Western blot analysis of UCP1 protein levels in the iWAT. (d-e) Representative images and muscle cell size analysis of H&E-stained muscle sections. (f) Relative mRNA levels of genes expressed in the muscle. (g-h) Representative images and lipid droplet size analysis of H&E-stained liver sections. (i) Gene expression of PCK1, G6PC and FOXO1 in the liver tissue. **p<0.01, ***p<0.001 by two-tailed Student's t-tests.

(**Response Letter Fig.11h**), which is consistent with the significantly elevated blood glucose and impaired glucose tolerance (**manuscript Fig.1h,i**) and insulin resistance (**manuscript Fig. 1j,k**) in HFD-fed AMPK IKO mice. This result suggested that intestinal AMPK plays a key role in regulating hepatic gluconeogenesis.

2. Lines 56-57 mention that food intake on HFD is similar between the knockout and the control mice, but food intake data is not shown. Please show these data. Also, for the other experiments throughout the paper, if food intake data were collected, it would be good to include them in the paper.

Response Letter Figure 12. Food intake of DIO-AMPK Control mice and IKO mice (a), and metformin-treated AMPK control mice (b) and AMPK IKO mice (c). Values are means \pm s.e.m.

As suggested, we included the data about food intake of HFD-fed, as well as metformin-treated AMPK control mice and AMPK IKO mice (**Response Letter Fig. 12**) into the manuscript as new **supplementary Fig. 1o** and **Supplementary Fig. 6a-b**, respectively.

2. *Figure 2: The number of mice is relatively low for the experiments in Figure 2 (n=3-4). Please indicate if these studies have been repeated in other cohorts. These experiments should be repeated to increase the number of mice. This is also the case for Supplementary Figures 2 and 3 and parts of Supplementary Figure 1 with n=4.*

For the Figure 2 and supplementary Figures 1-3 with n=3 or 4, we repeated in other cohorts of mice and updated these figures with final results in the revised manuscript.

- *Since mice are coprophagic, it is also important to note whether these mice were caged together or not and whether they came from the same litter.*

We have several pairs of breeders that derived from the same original litter to get enough number of mice for experiments. AMPK^{fl/fl} and IKO mice were caged separately with 5 mice per cage in a same room.

- *The method for Figure 2b is not explained anywhere. This should be described either in the figure legend or the Methods section.*

We apologize for our negligence. For the method of Figure 2b, we used the PICRUST (Phylogenetic Investigation of Communities by Reconstruction of Unobserved States) software package, based on the Clusters of Orthologous Groups (COGs) database, to do the analysis.

- *For the FMT experiment, how many mice were used as donors, and how were the donor mice selected? Were feces from all the donor mice mixed together, and all the recipient mice received the same fecal slurry? Or did each recipient mouse receive feces from a different donor?*

For the FMT experiment, we randomly selected six 10-12-week-old male mice maintained on normal chow diet feeding as donors. The fresh feces collected from all the donor mice were mixed together and all the recipient mice received the same fecal slurry.

- *In the figure legend for Figure 2k, please provide a more detailed description of the methylglyoxal injection (e.g., dose and frequency of injection).*

We apologize for the missing information. In Figure 2k, the mice were injected intraperitoneally (i.p.) with methylglyoxal (50 mg/kg) once daily for 14 days. We have added this information into the figure legend.

3- *Could the authors discuss more about the taxonomic composition of the gut microbiome? These data are shown in Supplementary Figure 2 but there is a lack of description about these taxa and how these findings relate to previous literature.*

As suggested, we have added the result description and relative discussion into the manuscript (lines 112-117) as follows: We examined the changes of gut microbiota at the family level and found that the relative abundance of families *Muribaculaceae*, *Prevotellaceae* and *Mycoplasmataceae* were significantly higher, while the relative abundance of family *Lachnospiraceae* was significantly lower in AMPK IKO mice (**Supplementary Fig. 2d**). It has been reported that higher levels of families *Muribaculaceae* and *Prevotellaceae* and lower level of *Lachnospiraceae* family are correlated with colitis in mice (Anshu, Babbar et al. 2019, Osaka et al. 2017, Bunker et al. 2015, Aida Iljazovic et al. 2021)(Frank DN et al. 2007). Further, we observed significant lower level of genus *Candidatus Arthromitus* that is a commensal bacterium necessary for inducing the postnatal maturation of

homeostatic innate and adaptive immune responses in the mouse gut (Alexander Bolotin et al. 2014). These results support the role of gut microbiota in AMPK-regulated intestinal barrier. Interestingly, the relative level of *Lachnospiraceae* family has been reported to be induced by cold stimulation in WT mice (Baoguo et al, 2019), suggesting that *Lachnospiraceae* family maybe involved in the intestinal AMPK-mediated energy expenditure.

References:

Babbar A et al. The Compromised Mucosal Immune System of $\beta 7$ Integrin-Deficient Mice Has Only Minor Effects on the Fecal Microbiota in Homeostasis. *Frontiers in Microbiology* **10**, 2284 (2019).

Baoguo Li et al. Microbiota Depletion Impairs Thermogenesis of Brown Adipose Tissue and Browning of White Adipose Tissue, *Cell Reports* **26**, 2720-2737 (2019).

Bolotin A et al. Genome Sequence of "Candidatus Arthromitus" sp. Strain SFB-Mouse-NL, a Commensal Bacterium with a Key Role in Postnatal Maturation of Gut Immune Functions. *Genome Announc* **2**, e00705-14 (2014).

Bunker JJ et al. Innate and Adaptive Humoral Responses Coat Distinct Commensal Bacteria with Immunoglobulin A. *Immunity* **43**, 541-553 (2015).

Frank DN et al. Molecular-phylogenetic characterization of microbial community imbalances in human inflammatory bowel diseases. *Proc Natl Acad Sci U S A* **104**, 13780-13785 (2007).

Iljazovic, A., Roy, U., Gálvez, E.J.C. et al. Perturbation of the gut microbiome by *Prevotella* spp. enhances host susceptibility to mucosal inflammation. *Mucosal Immunol* **14**, 113–124 (2021).

Osaka T et al. Meta-analysis of fecal microbiota and metabolites in experimental colitic mice during the inflammatory and healing phases. *Nutrients* **9**,1329 (2017).

Wehkamp J et al. Defensins and other antimicrobial peptides in inflammatory bowel disease. *Curr Opin Gastroenterol* **23**, 370-378 (2007).

4- For Figures 2c-f, 2i, and Supplementary Figure 3, how long after FMT these data were taken (how many days or weeks post-FMT)? Were the recipient mice fed with chow diet? These details are missing in the manuscript.

We apologize for the missing information. The data in Figures 2c-f, 2i, and Supplementary Figure 3 were taken 4 weeks after FMT. The recipient mice were fed chow diet. We have added this information into the revised manuscript.

5. The conclusions stated in lines 94-95 and lines 102-103 are not supported by the data. Here, the authors mention that increased methylglyoxal levels in AMPK-IKO mice are due to increased production of this metabolite by the gut microbiota. However, methylglyoxal is also produced by the host, as mentioned in lines 89-90. There is no evidence in the manuscript that the gut microbiome produces methylglyoxal in the AMPK-IKO mice. Intestinal AMPK knockout may lead to increased methylglyoxal production by the host. It is crucial to establish the origin of these metabolites highlighted in Figure 2g since one of the central claims in the paper is that gut microbiota metabolites mediate crosstalk between intestinal AMPK and BAT thermogenic regulation.

Thank you for the insightful comments. To address this issue, we examined methylglyoxal levels in the feces of AMPK control mice and IKO mice as well as FMT recipient

Response Letter Figure 13. Methylglyoxal levels in the fecal samples from AMPK IKO mice (a) and FMT recipient mice (b) fed ND chow. FMT-CC, FMT from AMPK control mice to WT mice; FMT-KC, FMT from AMPK-IKO mice to WT mice, *p<0.05, ** p<0.01 by two-tailed Student's t-tests.

mice. The methylglyoxal levels in the feces of AMPK IKO mice were significantly higher than AMPK control mice (**Response Letter Fig. 13a, Supplementary Fig. 3f**). In addition, the MG levels in the feces of the FMT recipient mice that received the fecal microbiome from AMPK IKO mice (FMT-KC) were also significantly higher than that in recipient mice that received the fecal microbiome from AMPK control mice (FMT-CC) (**Response Letter Fig. 13b, Supplementary Fig. 3g**). These results suggest that gut microbiota altered by AMPK IKO contributed significantly to the increased serum methylglyoxal levels.

6. *Figure 3: The conclusion (lines 123-125) is not strongly supported by the data. This statement says that AMPK modulates the gut microbiome through Reg3 γ . However, the results only show that intestinal AMPK regulates Reg3, but no data is showing that Reg3 regulates the gut microbiome or their metabolites (e.g. methylglyoxal). There may be no causal link between Reg3 expression in these mice and the gut microbiome.*

It's well-known that AMPs are epithelial innate antibiotic effector molecules with important roles in regulating the gut microbial community (Ostaff et al 2013; Natividad et al. 2013; Loonen et al. 2014; Cash et al. 2006; Morampudi et al. 2016), our results demonstrated that intestinal AMPK regulates AMPs, therefore, intestinal AMPK may modulate GM profile through regulating the levels of AMPs. We apologize for our overstated conclusion. We have revised the conclusion as: Taken together, these findings suggest that intestinal AMPK may modulate gut microbiota composition through regulation of AMPs, such as Reg3 γ , ...”.

References:

Cash HL, Whitham CV, Behrendt CL, Hooper LV. Symbiotic Bacteria Direct Expression of an Intestinal Bactericidal Lectin. *Science (New York, NY)* **313**, 1126-1130 (2006).

Loonen LM, et al. REG3 γ -deficient mice have altered mucus distribution and increased mucosal inflammatory responses to the microbiota and enteric pathogens in the ileum. *Mucosal Immunol* **7**, 939-947 (2014).

Morampudi V, et al. The goblet cell-derived mediator RELM- β drives spontaneous colitis in Muc2-deficient mice by promoting commensal microbial dysbiosis. *Mucosal immunology* **9**, 1218-1233 (2016).

Natividad JM, et al. Differential induction of antimicrobial REGIII by the intestinal microbiota and *Bifidobacterium breve* NCC2950. *Applied and environmental microbiology* **79**, 7745-7754 (2013).

Ostaff MJ, et al. Antimicrobial peptides and gut microbiota in homeostasis and pathology. *EMBO molecular medicine* **5**, 1465-1483 (2013).

7. *Figure 4: These results are quite striking, and it would be helpful to describe these in the context of previous findings. Other studies have shown the importance of liver AMPK in mediating metformin's actions. The authors should provide a discussion comparing the results of this study with these previously published results. The discussion may include the extent to which intestinal AMPK, relative to AMPK in other tissues, might mediate metformin's effects.*

Thank you for the insightful suggestion. As suggested, we have added the following discussion in the manuscript (line 188-202):

“Previous reports have demonstrated that hepatic AMPK is required to mediate the anti-hyperglycemic effects of metformin (Zhou et al. 2001; Shaw et al. 2005). However, there are also evidence showing that metformin inhibits hepatic gluconeogenesis in mice is independently of the LKB1/AMPK pathway (Fullerton et al. 2013; Madiraju et al. 2014) in the liver. Therefore, both AMPK-dependent and independent pathways in the liver are involved in the therapeutic effects of metformin. On the other hand, accumulating evidence indicate that gut may be another critical site for metformin to exhibit its therapeutic

effects. For instance, the peak of metformin concentrations in the jejunum has been reported at 30–300 times greater than plasma concentrations (Bailey et al. 2008), indicating that most of the metformin administered by gavage distributes to the intestine. Moreover, delayed-release metformin, which is formulated to deliver the drug to target the lower bowel, was more effective at lowering fasting plasma glucose than currently available metformin (Buse et al. 2016). In the gut, metformin can stimulate glucagon-like peptide-1 (GLP-1) release (Duca et al. 2015). Metformin is also involved in modulating gut microbiota that can regulate metabolism (Forslund et al. 2015; Wu et al. 2017). Importantly, when small intestinal AMPK was virally knocked-down in diabetic rodents, the glucose-lowering ability of acute metformin treatment was diminished by about 50%. Together, these results suggest a critical contribution of intestinal AMPK to the therapeutic effect of metformin, and might also explain the less extent to which hepatic AMPK deficiency impaired the therapeutic effects of metformin (Fullerton et al. 2013; Madiraju et al. 2014) than that caused by intestinal AMPK knockout (Fig. 4a-k).”

References:

- Bailey CJ, et al. *Metformin and the intestine. Diabetologia* **51**, 1552-1553 (2008).
- Buse JB, et al. *The Primary Glucose-Lowering Effect of Metformin Resides in the Gut, Not the Circulation: Results From Short-term Pharmacokinetic and 12-Week Dose-Ranging Studies. Diabetes care* **39**, 198-205 (2016).
- Duca FA, et al. *Metformin activates a duodenal Ampk-dependent pathway to lower hepatic glucose production in rats. Nature medicine* **21**, 506-511 (2015).
- Forslund K, et al. *Disentangling type 2 diabetes and metformin treatment signatures in the human gut microbiota. Nature* **528**, 262-266 (2015).
- Fullerton MD, et al. *Single phosphorylation sites in Acc1 and Acc2 regulate lipid homeostasis and the insulin-sensitizing effects of metformin. Nature medicine* **19**, 1649-1654 (2013).
- Madiraju AK, et al. *Metformin suppresses gluconeogenesis by inhibiting mitochondrial glycerophosphate dehydrogenase. Nature* **510**, 542-546 (2014).
- Shaw RJ, et al. *The kinase LKB1 mediates glucose homeostasis in liver and therapeutic effects of metformin. Science* **310**, 1642-1646 (2005).
- Wu H, et al. *Metformin alters the gut microbiome of individuals with treatment-naive type 2 diabetes, contributing to the therapeutic effects of the drug. Nature medicine* **23**, 850-858 (2017).
- Zhou G, et al. *Role of AMP-activated protein kinase in mechanism of metformin action. The Journal of clinical investigation* **108**, 1167-1174 (2001).

8- The authors could elaborate more on some of the details and relate the microbiome findings with previous literature. Are the metformin-induced microbiome alterations shown in this study similar to those shown in previous studies? If the metformin-treated mice's microbiome profiles are different from previous findings, what are some possible explanations?

As suggested, we have added the following discussion in the manuscript (line 211-216):

“...we found that metformin treatment significantly altered the relative amount of several bacterial genera in AMPK^{fl/fl} (**manuscript Fig. 4m, Supplementary Fig. 6g-h**), such as the increased levels of *Bacteroides* (Chen et al. 2018), *Parabacteroides* (Lee et al. 2018; Ryan et al. 2020; Barendolts et al. 2018), *Alistipes* (Qin et al. 2012; Larsen et al. 2010), *Akkermansia* (Shin et al. 2014; Bornstein et al. 2017), *Parasutterella* (Zhang et al. 2015) and *Klebsiella* (Zhang et al. 2015; Hiel et al. 2020), and decreased levels of *Blautia* (Shin et al. 2014), *Ruminiclostridium* (Elbere et al. 2018), *Lachnospiraceae_NK4A136_group* (Cui et al. 2019), *Oscillibacter* (Tong et al. 2018), *Bilophila* (Karlsson et al. 2013), *Desulfovibrio* (Cui et al. 2019; Ryan et al. 2020), *Streptococcus* (Ji, Wang et al. 2019), *Roseburia* (Hiel et al. 2020; Zhang et al. 2015), *Lachnoclostridium* (Ryan et al. 2020) and *Anaerotruncus* (Shin et al. 2014) that are consistent with previous reports. Interestingly, most of them are changed in an

intestinal AMPK-dependent manner (**manuscript Fig. 4m**). These results suggest that the metformin-controlled alteration of gut microbiota depends on intestinal AMPK. “

References:

- Barengolts E, et al. Gut microbiota varies by opioid use, circulating leptin and oxytocin in African American men with diabetes and high burden of chronic disease. *PLoS one* **13**, e0194171 (2018).
- Bornstein S, et al. Metformin Affects Cortical Bone Mass and Marrow Adiposity in Diet-Induced Obesity in Male Mice. *Endocrinology* **158**, 3369-3385 (2017).
- Chen C, et al. Modulation of gut microbiota by mulberry fruit polysaccharide treatment of obese diabetic db/db mice. *Food & function* **9**, 3732-3742 (2018).
- Cui HX, et al. A Purified Anthraquinone-Glycoside Preparation From Rhubarb Ameliorates Type 2 Diabetes Mellitus by Modulating the Gut Microbiota and Reducing Inflammation. *Frontiers in microbiology* **10**, 1423 (2019).
- Depommier C et al. Pasteurized *Akkermansia muciniphila* increases whole-body energy expenditure and fecal energy excretion in diet-induced obese mice. *Gut Microbes* **11**, 1231-1245 (2020).
- Elbere I, et al. Association of metformin administration with gut microbiome dysbiosis in healthy volunteers. *PLoS one* **13**, e0204317 (2018).
- Hiel S, et al. Link between gut microbiota and health outcomes in inulin -treated obese patients: Lessons from the Food4Gut multicenter randomized placebo-controlled trial. *Clinical nutrition (Edinburgh, Scotland)* **39**, 3618-3628 (2020).
- Ji S, et al. Effect of Metformin on Short-Term High-Fat Diet-Induced Weight Gain and Anxiety-Like Behavior and the Gut Microbiota. *Frontiers in endocrinology* **10**, 704 (2019).
- Karlsson FH, et al. Gut metagenome in European women with normal, impaired and diabetic glucose control. *Nature* **498**, 99-103 (2013).
- Larsen N, et al. Gut microbiota in human adults with type 2 diabetes differs from non-diabetic adults. *PLoS one* **5**, e9085 (2010).
- Lee H, et al. Modulation of the gut microbiota by metformin improves metabolic profiles in aged obese mice. *Gut microbes* **9**, 155-165 (2018).
- Qin J, et al. A metagenome-wide association study of gut microbiota in type 2 diabetes. *Nature* **490**, 55-60 (2012).
- Ryan PM, et al. Metformin and Dipeptidyl Peptidase-4 Inhibitor Differentially Modulate the Intestinal Microbiota and Plasma Metabolome of Metabolically Dysfunctional Mice. *Canadian journal of diabetes* **44**, 146-155 e142 (2020).
- Shin NR, et al. An increase in the *Akkermansia* spp. population induced by metformin treatment improves glucose homeostasis in diet-induced obese mice. *Gut* **63**, 727-735 (2014).
- Tong X, et al. Structural Alteration of Gut Microbiota during the Amelioration of Human Type 2 Diabetes with Hyperlipidemia by Metformin and a Traditional Chinese Herbal Formula: a Multicenter, Randomized, Open Label Clinical Trial. *mBio* **9**, e02392-17 (2018).
- Zhang X, et al. Modulation of gut microbiota by berberine and metformin during the treatment of high-fat diet-induced obesity in rats. *Scientific reports* **5**, 14405 (2015).

9. In general, bar charts would be clearer if represented as box plots with individual data points shown, so that the distribution of the data can be seen. Whenever possible, quantify the western blots and histology images.

Thank you for the suggestion. We have updated the figures to box plots and quantified the western blots and histology images in the manuscript.

10. The manuscript has a few grammatical errors and typos, including:

- For Supplementary Figure 2 legend, the last sentence is missing a period before “Values”.
- Line 79 is missing “that” before “received FMT”.
- Figure 2f legend: remove “mice” before “recipient mice”.

- Line 87: “funcions” should be “functions” instead.
- Supplementary Figure 1: The figure legend describes figures s-t, but there is no figure s or t.

We apologize for the grammatical errors and typos. We have corrected them in the manuscript.

Reviewer #3 (Remarks to the Author):

Zhang et al. reported ‘intestinal AMPK modulation of microbiota mediates cross-talk with brown fat to control thermogenesis’. This study is timely, well-written, and well-executed.

The authors genetically deleted AMPK alpha 1 in the IECs. AMPK-IKO mice on chow diet had marked adipocyte hypertrophy and increased lipid droplets as well as down-regulation of thermogenic gene program, and altered genes in mitochondrial function, fatty acid oxidation, and lipolytic pathways, together with an inhibition on energy expenditure. On a HFD, AMPK-IKO mice gained more weight, developed impaired glucose and insulin tolerance. Further, changes in gut microbiome were also detected in AMPK-IKO mice on chow diet and FMT of the gut microbiome obtained from AMPK-IKO replicated the metabolic phenotypes in AMPK-IKO mice. Next, the authors discovered that AMPK-IKO mice had higher levels of microbiota-derived methylglyoxal and the effect was replicated in FMT of the gut microbiome from AMPK-IKO mice. More importantly, direct administration of methylglyoxal per se replicated the phenotypes of AMPK-IKO mice. The Authors then reported AMPK-IKO mice had reduced intestinal Reg3γ, and that metformin increased intestinal Reg3γ expression in control but not AMPK-IKO mice. Interestingly, people with obesity and T2D also had reduced duodenal AMPK activity as well as Reg3γ expression. Then the authors found that metformin exerted benefits (i.e., weight loss and improvement on glucose tolerance) as well as changes in gut microbiome were attenuated in AMPK-IKO mice. I only have one question: Re the glucose tolerance observation reported in Figure 1h and Figure 4b, is the effect dependent on weight changes? Would the authors still observe the same trend if weight changes were controlled for?

We appreciate this reviewer’s positive comments. As shown in the **Response Letter Fig. 14**, although weight changes did affect the glucose tolerance results, we observed a similar trend when the body weight changes were controlled for the glucose tolerance in the comparison between AMPK control mice and IKO mice, as well as that between metformin-treated mice and vehicle-treated mice.

Response Letter Figure 14. (a) Glucose tolerance test relative to the body weight at 10 weeks of HFD feeding (n=5-7). (b) Glucose tolerance test results at 8 weeks of metformin administration (n=7). Values are mean ± SEM. *p<0.5, **p<0.01 by two-way ANOVA with Tukey’s post-hoc tests.

Reviewer #4 (Remarks to the Author):

The goal of this manuscript by Zhang et al is to demonstrate that intestinal AMPK and brown adipose tissue communicate via gut microbiota-derived metabolites. Overall, this is a robust manuscript that reports interesting communication between intestine and brown adipose tissue. Nevertheless, I have

the following significant concerns with regards to metabolomics experimentation and metabolomics data interpretation:

We appreciate the positive and insightful comments from this reviewer. Please find below our point-by-point response to individual comment.

1. Methods leading to Figure 2b are missing.

We apologize for the missing information. We have added the detailed method in the Method section of the manuscript (lines 308-310).

2. Metabolomics methods are insufficient. Missing information includes: metabolite extraction method, all LC-MS parameters and all data analysis parameters. Authors are referred to the Metabolomics Standards Initiative (PMC3772505) for a comprehensive list of the information that needs to be provided.

Thank you for the suggestion. We have added the detailed method including the required information in the Method section of manuscript (lines 328-357).

3. Authors must provide evidence to support the metabolite annotations provided in Figure 2g. Acceptable evidence would be MS2 spectral matching to reference spectrum, or matched extracted ion chromatograms compared to a pure standard. This is particularly important given multiple unlikely metabolite annotations in Figure 2g, such as prilocaine, gaboxadol, coumarin, exalamide, perillyl alcohol, pargyline and lavandulol.

The evidence to support the metabolite annotations were provided in the following **Response Letter Fig.15**, which showed the MS2 spectral matching to reference spectrum.

The metabolomics study was conducted using the untargeted metabolomics strategy. This strategy was reported to have possibility to identify the metabolites from plants and medicines. We have removed those unlikely metabolites as indicated in the revised manuscript.

Response Letter Figure 15. The MS2 spectral matching to reference spectrum for the metabolites in Original Figure 2g.

4. Given these dubious annotations, confirming methylglyoxal annotation to the standards described above is essential, particularly since it is the basis for multiple subsequent experiments.

We agree that it's essential to confirm methylglyoxal annotation to the standard. We performed liquid chromatography-tandem mass spectrometry (LC-MS/MS) to examine the MG levels according to the previous study (Rabbani et al. 2014) that established a method of measuring methylglyoxal by stable isotopic dilution analysis LC-MS/MS with corroborative prediction in physiological samples. In this method, MG can be derivatized with 1,2-diaminobenzene (DB), resulting in an adduct that can be detected using LC-MS/MS. Quantification is achieved by stable isotopic dilution analysis with [¹³C₃] MG. **Response Letter Figure 16a** showed the standard curve, and **Response Letter Figure 16b** showed the sample curve matching with the standard solution curve. These results demonstrated that this method is reliable for the measurement of MG levels. Then we used this method to measure the serum and fecal MG levels in AMPK IKO mice and the FMT recipient mice. As shown in the **Response Letter Figure 16c-d**, the MG levels in the serum and feces of AMPK IKO mice were significantly increased compared to that of the AMPK control mice. In addition, the MG levels in the feces of the FMT recipient mice that received the fecal microbiome from AMPK IKO mice (FMT-KC) were also significantly higher than that in recipient mice that received the fecal microbiome from AMPK control mice (FMT-CC) (**Response Letter Fig. 16e, Supplementary Fig. 3g**). These results suggest that gut microbiota altered by AMPK IKO may contribute significantly to the increased circulating methylglyoxal levels in AMPK-IKO mice. We have added the detailed method in the Method section (lines 359-386).

Response Letter Figure 16. The analysis of methylglyoxal using by LC-MS/MS. (a) The standard curve (b) the sample curve (below) matched with the standard curve (upper). (c-e) MG levels in the serum of DIO mice (c), the fecal samples of ND-fed mice (d) and FMT recipient mice (e). * $p < 0.05$, ** $p < 0.01$ by two-tailed Student's t-tests.

Reference:

Rabbani N, et al. Measurement of methylglyoxal by stable isotopic dilution analysis LC-MS/MS with corroborative prediction in physiological samples. *Nat Protoc* 9, 1969-1979 (2014).

5. I would recommend that the authors consider publicly depositing their metabolomics data, as is standard for sequencing data.

Thank you for the suggestion. We will deposit our metabolomics data publicly.

6. A central tenet of this manuscript is that intestine-AMPK communication is via microbiota metabolites, specifically methylglyoxal. However, this metabolite is ubiquitous and can also be produced by human cells. Elevated methylglyoxal in cecum contents rather than serum would be a more convincing indication of microbiota source than serum analyses, as would indication that it is produced by bacteria cultured from IKO microbiota.

Thank you for the insightful suggestion. We examined methylglyoxal levels in the feces of AMPK IKO mice as well as the FMT recipient mice. The methylglyoxal levels in the feces of AMPK IKO mice were significantly higher than AMPK control mice (**Response Letter Fig. 17a, Supplementary Fig. 3f**). Also, the methylglyoxal levels in the feces of the FMT recipient mice that received fecal microbiome from AMPK IKO mice (FMT-KC) were significantly higher than that in the recipient mice that received fecal microbiome from AMPK control mice (FMT-CC) (**Response Letter Fig. 17b, Supplementary Fig. 3g**). These results suggest that gut microbiota altered by AMPK IKO may contribute significantly to the increased serum methylglyoxal levels in AMPK-IKO mice.

7. Likewise, confirming that increased serum methylglyoxal is a cause of the adipose tissue phenotype requires confirmation of elevated methylglyoxal in BAT. Otherwise, altered BAT phenotype could be an indirect effect.

We detected the methylglyoxal levels in the BAT of AMPK control mice and AMPK IKO mice using LC-MS/MS method. As shown in the **Response Letter Fig. 18**, we observed higher methylglyoxal levels in BAT of the AMPK IKO mice. Together with the increased serum and fecal methylglyoxal levels in AMPK IKO mice, and the *in vitro* data showing that methylglyoxal treatment suppressed UCP1 protein level in differentiated brown adipocyte HIB1B cells (**manuscript Fig. 2j**), these results suggest that the increasing serum methylglyoxal may be a cause of the BAT phenotype in mice.

8. Methylglyoxal is toxic. Authors should provide additional experiments to confirm that the effects of methylglyoxal on UCP1 levels are not merely a consequence of cell death.

Response Letter Figure 17. Methylglyoxal levels in the fecal samples in the AMPK mice (a) and FMT recipient mice (b) fed ND chow. FMT-CC, FMT from AMPK control mice to WT mice; FMT-KC, FMT from AMPK-IKO mice to WT mice, * $p < 0.05$, ** $p < 0.01$ by two-tailed Student's t-tests

Response Letter Figure 18. Methylglyoxal levels in the BAT tissue of AMPK control mice and AMPK IKO mice fed ND chow. * $p < 0.05$, by two-tailed Student's t-tests.

To test the cell toxicity of methylglyoxal, we performed MTT assay on methylglyoxal treated HIB1B cells. As shown in the **Response Letter Figure 19**, methylglyoxal did not affect the cell viability of HIB1B cells at the concentration up to 100 μM . In our study, methylglyoxal impaired brown adipocyte function from the concentration of only 0.05 μM (**manuscript Fig. 2j**). Moreover, we examined the protein levels of cell death marker Bcl2 and cleaved-caspase 3 in the BAT tissues, and didn't observe any changes between the vehicle- and methylglyoxal-treated groups (**Response Letter Fig. 20**). Taken together, these data indicate that the effects of methylglyoxal on UCP1 levels are not due to its toxicity.

Response Letter Figure 19. Cell viability test in the HIB1B cells treated with gradient concentrations of Methylglyoxal (MG). n=6. **p<0.01, ***p<0.001.

9. How do the authors explain the effects of methylglyoxal, given endogenous mechanisms for detoxifying it (e.g. glyoxalase), and the fact that methylglyoxal is rapidly bound to macromolecules?

MG is present in different forms. Only 1% of MG exists in a free unhydrated, monohydrated, or dihydrated form (Lo et al. 1994), and the major part is reversibly bound to proteins, peptides, and amino acids. The free and reversibly bound forms of MG are in dynamic equilibrium. Evidence have shown that fasting plasma MG concentrations are higher in T1D (Han et al. 2007) and T2D (Beisswenger et al. 1999; Kong et al. 2014; Lapolla et al. 2003; Odani et al. 1999; Scheijen and Schalkwijk 2014), and increase transiently during the postprandial period in both normoglycemic and diabetic individuals (Beisswenger et al. 2004; Maessen et al. 2015). Therefore, one explanation for the effects of methylglyoxal on BAT may be the presence of higher levels of unstable reversibly bound MG in the AMPK IKO mice compared to the AMPK control mice(Chaplen, Fahl, and Cameron 1998).

Response Letter Figure 20. Bcl2 and Cleaved-caspase3 protein expression in the BAT of WT mice after i.p. injections of methylglyoxal for 2 weeks.

References:

- Beisswenger P, et al. Prandial glucose regulation in the glucose triad: emerging evidence and insights. *Endocrine* **25**, 195-202 (2004).
- Beisswenger PJ, et al. Metformin reduces systemic methylglyoxal levels in type 2 diabetes. *Diabetes* **48**, 198-202 (1999).
- Chaplen FW, et al. Evidence of high levels of methylglyoxal in cultured Chinese hamster ovary cells. *Proc Natl Acad Sci U S A* **95**, 5533-5538 (1998).
- Han Y, et al. Plasma methylglyoxal and glyoxal are elevated and related to early membrane alteration in young, complication-free patients with Type 1 diabetes. *Molecular and cellular biochemistry* **305**, 123-131 (2007).
- Kong X, et al. Increased plasma levels of the methylglyoxal in patients with newly diagnosed type 2 diabetes 2. *Journal of diabetes* **6**, 535-540 (2014).
- Lapolla A, et al. Glyoxal and methylglyoxal levels in diabetic patients: quantitative determination by a new GC/MS method. *Clinical chemistry and laboratory medicine* **41**, 1166-1173 (2003).
- Lo TW, et al. Binding and modification of proteins by methylglyoxal under physiological conditions. A kinetic and mechanistic study with N alpha-acetylgarginine, N alpha-acetylcysteine, and N alpha-acetyllysine, and bovine serum albumin. *The Journal of biological chemistry* **269**, 32299-32305 (1994).
- Maessen DE, et al. Post-Glucose Load Plasma α -Dicarbonyl Concentrations Are Increased in Individuals With Impaired Glucose Metabolism and Type 2 Diabetes: The CODAM Study. *Diabetes care* **38**, 913-920 (2015).

Odani H, et al. Increase in three alpha,beta-dicarbonyl compound levels in human uremic plasma: specific in vivo determination of intermediates in advanced Maillard reaction. *Biochem Biophys Res Commun* **256**, 89-93 (1999).
 Scheijen JL, et al. Quantification of glyoxal, methylglyoxal and 3-deoxyglucosone in blood and plasma by ultra performance liquid chromatography tandem mass spectrometry: evaluation of blood specimen. *Clinical chemistry and laboratory medicine* **52**, 85-91 (2014).

10. Authors only describe intestinal epithelial cell isolation from the small intestine (lines 265-271). Given this manuscript's emphasis on AMPK-microbiota interaction, it is essential to confirm loss of AMPK in the KO mice in the cecum and large intestine as well as the small intestine.

In addition to the jejunum (Supplementary Fig. 1a), we have also measured the AMPK α 1 expression in the epithelial cells isolated from ileum and colon of the AMPK IKO mice and confirmed the loss of AMPK in the IKO mice (**Response Letter Fig. 21**). For the cecum tissue, we isolated the epithelial cells, but the protein expression is too low to be detected.

Response Letter Figure 21. AMPK α 1 protein level in the colon (left) and ileum (right) tissues of the AMPK IKO mice.

11. Though supplementary figure S1m and S1n data is marked as statistically significance, I question the biological validity of these findings, given the small magnitude of change.

The UCP1/BAT system has evolved to maintain body temperature (Kozak LP et al. 2010) as the primary role in body, whereas, leading to weight loss is secondary to the primary role (Kozak LP et al. 2010, Kozak LP et al. 2014). For instance, mice deficient in UCP1 do not have increased susceptibility to obesity (N.J. Rothwell et al. 1979). These may provide an explanation for the small magnitude of changes in Fig. S1m and S1n. Perhaps the main function of intestinal AMPK-mediated BAT is to produce heat for the body temperature maintenance as we discussed in the text.

References:

Kozak LP. Brown fat and the myth of diet-induced thermogenesis. *Cell Metab* **11**, 263-267 (2010).
 Kozak LP. Genetic variation in brown fat activity and body weight regulation in mice: Lessons for human studies, *Biochimica et Biophysica Acta (BBA) - Molecular Basis of Disease* **1842**, 370-376 (2014).
 Rothwell NJ, et al. A role for brown adipose tissue in diet-induced thermogenesis. *Nature* **281**,31-35. (1979).

12. Minor comments:

- 1) Figure 2j: what is the EC50 of methylglyoxal with regards to UCP1 expression in HIB1B cells?
- 2) Lines 138-139: "We found that metformin reduced lipid accumulation and UCP1 level in the BAT of AMPK fl/fl , but not AMPK-IKO mice". In fact, metformin appears to increase UCP1 in control mice (Figure 4j). Is this a typo/misphrasing?
- 3) Densitometry should be provided for Figure 4j, where differences between groups are harder to see (especially in IKO mice).
- 4) Lines 99-100: "Furthermore, we show that oral gavage of mice with methylglyoxal reduced UCP1 expression in BAT" but methods indicate "mice were treated with intraperitoneal (i.p.) injections of methylglyoxal (50 mg/kg) once daily for 14 days." (lines 202-203). Which route was actually used?

Response Letter Figure 22. UCP1 protein expression in the HIB1B cells treated with gradient concentrations of Methylglyoxal.

1). As suggested, we examined the effect of the different concentration of methylglyoxal on the UCP1 expression in

HIB1B cells. As shown in the **Response Letter Figure 22**, the EC50 of methylglyoxal with regards to UCP1 expression in HIB1B cells is about 0.01 μ M.

2). We apologize for the misphrasing. We have corrected this as “We found that metformin reduced lipid accumulation and increased UCP1 level in the BAT of AMPK^{fl/fl}, but not AMPK-IKO mice” in the manuscript.

3). Thank you for the suggestion. We have calculated the Densitometry for the Figure 4 and added the data as **Supplementary Fig. 6f** in the revised manuscript.

4). We apologize for the misphrasing. For the in vivo study, mice were treated with intraperitoneal (i.p.) injections of methylglyoxal (50 mg/kg) once daily for 14 days. We have corrected it in the manuscript.

Reviewers' Comments:

Reviewer #1:

Remarks to the Author:

In this revised manuscript the authors have added considerable new experimental data to address the reviewer's concerns and is substantially improved.

Major points:

- Supplementary Figure 1: it remains unclear if the AMPK band detected in this figure corresponds to AMPKalpha1 or both AMPKalpha1 and AMPKalpha2 as the M&M section indicates the use of a pan AMPKalpha antibody (Cell Signalling #2532 detecting both AMPKalpha1 and alpha2). The expression patterns of AMPKalpha subunits are quite different. While there are comparable amounts of AMPKalpha subunits in the heart, kidney, and liver, the amount of AMPKalpha1 is very low in cellular lysates from skeletal muscle. A control showing AMPKalpha1 expression in skeletal muscle is required to confirm the specificity of these antibodies. In addition, it would be nice to include in this figure, the western blot analysis for AMPKalpha2 expression in these tissues and also IEC (as shown in Response Letter Figure 3). Could the authors indicate the source for AMPKalpha1 and AMPKalpha2 antibodies used in this study?

Minor Points:

- Supplementary Figure 1: the upper blot showing AMPKalpha1 expression should be labelled « AMPKalpha1 » as indicated in the legend figure.

- line 87: correct lipolysis
- line 100: correct observation
- Figure 4 legend: replace (i) by (k) and (j) by (l).

Reviewer #2:

Remarks to the Author:

Thank you for providing comprehensive responses to the previous comments. All the concerns have been fully addressed. This paper has now merited a publication in Nature Communications.

Reviewer #3:

Remarks to the Author:

Thank you for addressing the weight-independent effect of AMPK IKO mice as well as the effect of metformin on glucose control in Response Letter Fig. 14. After re-reading the revised manuscript, I have the following minor requests.

(1) The authors should add the weight-independent gluco-regulatory effect of AMPK IKO mice to the text and as a supplementary figure.

(2) The authors addressed the weight-independent gluco-regulatory effect of metformin in wild-type mice. But did the authors (I am sorry if I did not make this request clearer before) have data evaluating whether metformin still fails to improve glucose tolerance in AMPK IKO mice independent of weight changes? In other words, does AMPK IKO disrupt the ability of metformin to increase glucose tolerance 'independent of weight changes'? The authors are recommended to add the weight-independent gluco-regulatory effect of metformin to the text and as a supplementary figure as well as any additional data showing whether or not metformin fails to increase glucose tolerance in AMPK IKO independent of weight changes.

(3) Finally, the authors are recommended to incorporate their respective findings; (i) AMPK IKO mice also gained weight and developed impaired glucose and insulin tolerance, and (ii) metformin failed to lower weight and increase glucose tolerance in AMPK IKO mice. in the body of the

Abstract.

Reviewer #4:

Remarks to the Author:

Most of my comments have been satisfactorily addressed. Remaining issues are as follows:

Point 2: XCMS peak deconvolution, alignment and integration parameters are still missing (line 362). This is very important to provide for reproducibility, because the output feature table will be influenced by parameter choice.

Point 3: Multiple MS2 mirror plots in Response Letter Fig.15 indicate poor spectral matches (N- α -acetyl-L-arginine, methylglyoxal, nicotinate, ala-leu, N-carboxyethyl- γ -aminobutyric acid, glycerol, phenylglycine, erythro-4-hydroxyglutamic acid, met-tyr), which give me low confidence in the reported annotations. The poor spectral match for methylglyoxal is especially worrisome since so much of the authors' subsequent analysis relies on this match. Were the library MS2 spectra acquired at the same collision energy as the reference spectra? This issue is partially mitigated by the chromatograms in Response Letter Figure 16b. For the remaining annotations with poor spectral matches, additional support should be provided in the form of retention time matching to authentic standards, or they should be removed from fig. 2g.

All the figures provided in the response to reviewers should be provided to readers as supplementary figures. They are important as support for the authors' findings.

Minor issues:

line 338: -40°C incubation. Is this a typo (meant -20°C or -80°C)? Please confirm.

line 347: typo: pH=9.75) ?

line 368: need to define TCA abbreviation

line 382: need to define TFA abbreviation

Point-by-Point Response to the Reviewers' Comments

Reviewer #1 (Remarks to the Author):

In this revised manuscript the authors have added considerable new experimental data to address the reviewer's concerns and is substantially improved.

We appreciate the positive and insightful comments from this reviewer. Please find below our point-by-point response to individual comment.

Major points:

- *Supplementary Figure 1: it remains unclear if the AMPK band detected in this figure corresponds to AMPK α 1 or both AMPK α 1 and AMPK α 2 as the M&M section indicates the use of a pan AMPK α antibody (Cell Signalling #2532 detecting both AMPK α 1 and α 2). The expression patterns of AMPK α subunits are quite different. While there are comparable amounts of AMPK α subunits in the heart, kidney, and liver, the amount of AMPK α 1 is very low in cellular lysates from skeletal muscle. A control showing AMPK α 1 expression in skeletal muscle is required to confirm the specificity of these antibodies. In addition, it would be nice to include in this figure, the western blot analysis for AMPK α 2 expression in these tissues and also IEC (as shown in Response Letter Figure 3). Could the authors indicate the source for AMPK α 1 and AMPK α 2 antibodies used in this study?*

Thank you for the suggestions. In order to clarify the AMPK α 1 and AMPK α 2 expression in different tissues and IEC in AMPK IKO mice, we ordered specific antibodies against AMPK α 1 (R&D, #AF3197) and AMPK α 2 (abcam, ab3760), respectively, and repeated the western blot analysis. As shown in the **Response letter Figure 1**, AMPK α 1 expression

Response letter Figure 1. Western blot analysis of AMPK α 1 and AMPK α 2 in various tissues of AMPK^{fl/fl} (Con) and AMPK-IKO mice. IEC, isolated epithelial cells; BAT, brown adipose tissue.

was only ablated in the IEC of AMPK IKO mice, but not in other tissues including the spleen, heart, lung, liver, kidney, muscle and BAT. Because the expression level of AMPK α 2 in IEC is much lower than that in other tissues, we can't observe the western blot bands of AMPK α 2 when the IEC samples were loaded and exposed together with other tissue samples. Thus, we exposed the membrane containing the IEC samples separately from the other tissue samples, and observed that the expression of AMPK α 2 in the IEC of AMPK IKO mice is

similar to that of the AMPK control mice. We have added this figure to the manuscript as **Supplementary Figure 1a**.

Minor Points:

- *Supplementary Figure 1: the upper blot showing AMPK α 1 expression should be labelled « AMPK α 1 » as indicated in the legend figure.*
- *line 87: correct lipolytic*
- *line 100: correct observation*
- *Figure 4 legend: replace (i) by (k) and (j) by (l).*

Thank you very much for your critical reading. We have revised/corrected them in the manuscript.

Reviewer #2 (Remarks to the Author):

Thank you for providing comprehensive responses to the previous comments. All the concerns have been fully addressed. This paper has now merited a publication in Nature Communications.

We appreciate the positive comments from this reviewer.

Reviewer #3 (Remarks to the Author):

Thank you for addressing the weight-independent effect of AMPK IKO mice as well as the effect of metformin on glucose control in Response Letter Fig. 14. After re-reading the revised manuscript, I have the following minor requests.

We appreciate the positive and insightful comments from this reviewer. Please find below our point-by-point response to individual comment.

(1) The authors should add the weight-independent glucoregulatory effect of AMPK IKO mice to the text and as a supplementary figure.

As suggested, we have added the weight-independent glucoregulatory effect of AMPK IKO mice to the text as **Supplementary Figure 1r**.

(2) The authors addressed the weight-independent glucoregulatory effect of metformin in wild-type mice. But did the authors (I am sorry if I did not make this request clearer before) have data evaluating whether metformin still fail to improve glucose tolerance in AMPK IKO mice independent of weight changes? In other words, does AMPK IKO disrupt the ability of metformin to increase glucose tolerance 'independent of weight changes'? The authors are recommended to add the weight-independent glucoregulatory effect of metformin to the text and as a supplementary figure as well as any additional data showing whether or not metformin fail to increase glucose tolerance in AMPK IKO independent of weight changes.

We observed that metformin failed to improve glucose tolerance in AMPK IKO mice independent of weight changes. We have added the weight-independent glucoregulatory effect of metformin, and the data that metformin failed to increase glucose tolerance in AMPK IKO independent of weight changes to the manuscript as **Supplementary Figure 9e-f**.

(3) Finally, the authors are recommended to incorporate their respective findings; (i) AMPK IKO mice also gained weight and developed impair glucose and insulin tolerance, and (ii) metformin failed to lower weight and increase glucose tolerance in AMPK IKO mice. in the body of the Abstract.

As suggested, we have added these findings in the body of the Abstract.

Reviewer #4 (Remarks to the Author):

Most of my comments have been satisfactorily addressed.

We appreciate the positive and insightful comments from this reviewer. Please find below our point-by-point response to individual comment.

Remaining issues are as follows:

Point 2: XCMS peak deconvolution, alignment and integration parameters are still missing (line 362). This is very important to provide for reproducibility, because the output feature table will be influenced by parameter choice.

As suggested, we have added the detailed information in the revised manuscript as the following: “The acquired MS data pretreatments including peak selection and grouping, retention time correction, second peak grouping, and isotopes and adducts annotation were performed as described previously ¹ with a few modifications. UHPLC-MS raw data files were converted into mzXML format using the “msconvert” program from ProteoWizard and then analyzed by the XCMS ² and CAMERA toolbox ³ with R statistical software. The CentWave algorithm in XCMS was used for peak detection. The parameter “peak-width” was set as (5, 20) in units of seconds, referring to the minimum and maximum peak widths for peak detection. The parameter “snthresh” is set as 3 for sensitive peak detection. For multiple UHPLC–MS data files, an ordered bijective interpolated warping (OBI-Warp) algorithm in XCMS was used for peak alignment ⁴. By using retention time and the m/z data pairs as the identifiers for each ion, we obtained ion intensities of each peak and generated a three dimensional matrix containing arbitrarily assigned peak indices (retention time-m/z pairs), ion intensities (variables) and sample names (observations).”

Reference:

1. Liu R, et al. Gut microbiome and serum metabolome alterations in obesity and after weight-loss intervention. *Nature Medicine* **23**, 859-868 (2017).
2. Tautenhahn R, Böttcher C, Neumann S. Highly sensitive feature detection for high resolution LC/MS. *BMC Bioinformatics* **9**, 504 (2008).
3. Kuhl C, Tautenhahn R, Böttcher C, Larson TR, Neumann S. CAMERA: an integrated strategy for compound spectra extraction and annotation of liquid chromatography/mass spectrometry data sets. *Analytical chemistry* **84**, 283-289 (2012).
4. Prince JT, Marcotte EM. Chromatographic alignment of ESI-LC-MS proteomics data sets by ordered bijective interpolated warping. *Analytical chemistry* **78**, 6140-6152 (2006).

Point 3: Multiple MS2 mirror plots in Response Letter Fig.15 indicate poor spectral matches (N- α -acetyl-L-arginine, methylglyoxal, nicotinate, ala-leu, N-carboxyethyl- γ -aminobutyric acid, glycerol, phenylglycine, erythro-4-hydroxyglutamic acid, met-tyr), which give me low confidence in the reported annotations. The poor spectral match for methylglyoxal is especially worrisome since so much of the authors' subsequent analysis relies on this match. Were the library MS2

spectra acquired at the same collision energy as the reference spectra? This issue is partially mitigated by the chromatograms in Response Letter Figure 16b. For the remaining annotations with poor spectral matches, additional support should be provided in the form of retention time matching to authentic standards, or they should be removed from fig. 2g.

The serum metabolomics was performed by a CRO service, and we have confirmed with the service company about the detailed information. Regarding the collision energy, not all the obtained library MS2 spectra are the same as the collision energy of the reference spectra, but most of them are similar. Because we focus on the serum metabolite methylglyoxal in this study, we therefore further established a LC-MS/MS method to confirm the methylglyoxal levels in our mouse models. For the remaining annotations with poor spectral matches, we will select the promising candidates to establish the LC-MS/MS methods in our future study. As suggested, we have removed them from Fig. 2g.

All the figures provided in the response to reviewers should be provided to readers as supplementary figures. They are important as support for the authors' findings.

Thanks for the suggestions, we have added the Response Letter figures in the manuscript as supplementary figures.

Minor issues:

line 338: -40°C incubation. Is this a typo (meant -20°C or -80°C)? Please confirm.

line 347: typo: pH=9.75) ?

line 368: need to define TCA abbreviation

line 382: need to define TFA abbreviation

Line 338: The incubation temperature is -40°C in order to precipitate protein.

Line 347: We have corrected the typo “pH=9.75)” to “(pH=9.75)”.

Line 368: We've added the full name for TCA as “trichloroacetic acid (TCA)”.

Line 382: We've added the full name for TFA as “trifluoroacetic acid (TFA)”.

Reviewers' Comments:

Reviewer #1:

Remarks to the Author:

The authors have done a commendable job in addressing AMPK α 1 and AMPK α 2 expression concerns. Western blot analysis clearly showed that both AMPK α 1 and AMPK α 2 isoforms are expressed IEC and that AMPK α 2 expression in IEC is not affected by AMPK α 1 deletion. Obviously, description of "an intestinal epithelium-specific AMPK-null (AMPK-IKO) mouse model" is inappropriate in the abstract as well as in the entire manuscript (text + figures + legends). Therefore, to avoid confusion for the readers, I would recommend to re-named the mouse model used in this study by indicating the specific deletion of AMPK α 1 isoform only (e.g., AMPK α 1-IKO). Similarly, AMPKfl/fl mice should be renamed as AMPK α 1fl/fl mice.

Based on this recommendation, I am supportive of publication.

Reviewer #3:

Remarks to the Author:

The authors have addressed my remaining comments.

Reviewer #4:

Remarks to the Author:

All my comments have been satisfactorily addressed.

REVIEWERS' COMMENTS

Reviewer #1 (Remarks to the Author):

The authors have done a commendable job in addressing AMPK α 1 and AMPK α 2 expression concerns. Western blot analysis clearly showed that both AMPK α 1 and AMPK α 2 isoforms are expressed IEC and that AMPK α 2 expression in IEC is not affected by AMPK α 1 deletion. Obviously, description of "an intestinal epithelium-specific AMPK-null (AMPK-IKO) mouse model" is inappropriate in the abstract as well as in the entire manuscript (text + figures + legends). Therefore, to avoid confusion for the readers, I would recommend to re-named the mouse model used in this study by indicating the specific deletion of AMPK α 1 isoform only (e.g., AMPK α 1-IKO). Similarly, AMPKfl/fl mice should be renamed as AMPK α 1fl/fl mice.

Based on this recommendation, I am supportive of publication.

We appreciate the positive and insightful comments from this reviewer. We have revised the descriptions in the manuscript as suggested.

Reviewer #3 (Remarks to the Author):

The authors have addressed my remaining comments.

We appreciate the positive comments from this reviewer.

Reviewer #4 (Remarks to the Author):

All my comments have been satisfactorily addressed.

We appreciate the positive comments from this reviewer.